# Vehicular Ad Hoc Networks Routing Strategies for Intelligent Transportation System

**Ishtiaq Wahid [1], Sadaf Tanvir [2], Masood Ahmad [1], Fasee Ullah [3,*], Ahmed S AlGhamdi [4], Murad Khan [5] and Sultan S. Alshamrani [6]**

[1] Department of Computer Science, Abdul Wali Khan University Mardan, Mardan 23200, Pakistan; ishtiaqwahid@awkum.edu.pk (I.W.); masood@awkum.edu.pk (M.A.)
[2] Department of Computing and Technology, Iqra University, Islamabad 44000, Pakistan; sadaf.tanvir@iqraisb.edu.pk
[3] Department of Computer Science & IT, Sarhad University of Science & IT, Peshawar 25120, Pakistan
[4] Department of Computer Engineering, Collage of Computers and Information Technology, Taif University, Taif 21944, Saudi Arabia; asjannah@tu.edu.sa
[5] Kuwait College of Science and Technology, Kuwait City 35001, Kuwait; m.khan@kcst.edu.kw
[6] Department of Information Technology, College of Computer and Information Technology, Taif University, Taif 21944, Saudi Arabia; susamash@tu.edu.sa
* Correspondence: faseekhan@gmail.com

**Abstract:** The upcoming models of vehicles will be able to communicate with each other and will thus be able to share and/or transfer information. A vehicular ad hoc network (VANET) is an application of this vehicular communication that leads to an intelligent transportation system (ITS). Vehicle-to-vehicle (V2V) and vehicle-to-infrastructure (V2I) are the two distinct types of vehicular ad hoc networks (VANET). V2V and V2I technologies are together known as V2X and are recently being tested. Continuous research to enhance routing considers different characteristics and exciting aspects of VANETs. The proposed schemes are classified based on the operational scenario. A survey of proposed routing schemes in the last eight years is presented to determine the design considerations and the approach used in every proposed system, along with their shortcomings. This survey will assist new scholars in this field to analyze existing state-of-the-art systems. The table at the end of each routing scheme shows the proposed routing scheme's simulation, routing, and scenario parameters. This paper also reviews VANET technology, its role in the intelligent transportation system, recent development in the field, and the timeline for implementation of the system.

**Keywords:** ITS; mobility; protocols; QoS; routing; VANET

## 1. Introduction

A vehicle ad hoc network has developed into an exciting but challenging area in which many new applications may find their place. Though the research in this area has been going on for a couple of decades, practical implementation at large-scale would take more time [1]. Nascent models in the automotive industry will manage to communicate with each other and exchange online information. Vehicular ad hoc network (VANET), as an on-board application of communication, is leading to an intelligent transportation system (ITS) [2]. A key factor for emerging ITS applications is all (V2X) communication, which allows vehicles to communicate with other vehicles, walkers, road infrastructure, and the Internet [3]. With V2X, vehicles are connected to each other and provide drivers with alerts and warnings about road conditions and hazards. In the near future, the vehicle will interact with its drivers and be connected to nearby vehicles. It will have awareness of its surroundings and road conditions. In the future, driving this way will be able to avoid heavy traffic congestion and road crashes and to guarantee safety on the roads.

Each vehicle node in V2V is a part of a mesh network that communicates messages, receives messages, and resends messages if essential. Three standards, IEEE 802.11p, SAE J2735/SAE J2945, and IEEE 1609, define the network architecture for this network, message packet data and a physical standard for Dedicated Short-Range Communication (DSRC). DSRC enables communications among vehicles, roadside units, and inter-vehicle (V2V) [4]. Vehicle sensors provide information about speed, braking, location, and direction of travel to the network.

Ambient components, like signal lights and sensors (installed at roadside assisting V2X), are acting as network nodes in the network. V2I provide support to vehicle/nodes to assist the network in informing vehicle drivers about signal light timing, road signs, and hazards. Figure 1 shows that V2X is a vehicle to everything technology in which V2I plays a role in warnings and alerts associated with the timing and priority of traffic signals. V2N, vehicles to network, informs the vehicles about real-time cloud services, traffic, routing, etc. V2V plays a role in safety systems to avoid collisions. Vehicle-to-pedestrian V2P gives safety warnings for pedestrians and cyclists. Autotalks is a company that offers V2X solutions. They have placed a banner on their website saying, "Wait until 2024". This statement shows that V2V will be a reality soon [5].

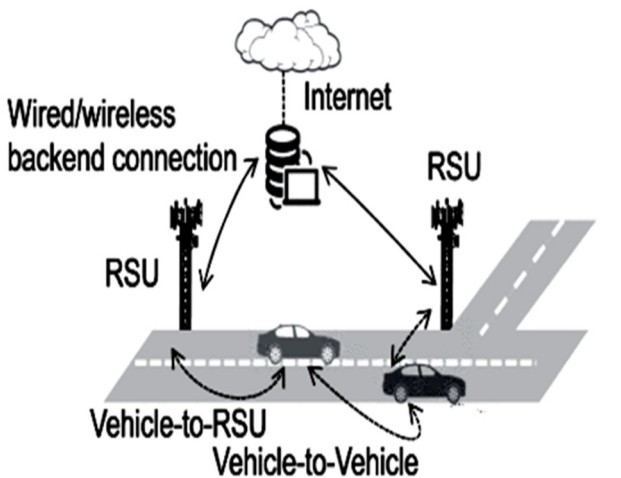

**Figure 1.** Vehicle to everything V2X technology [6].

The following section presents these routing protocol design considerations based on the approach proposed in each algorithm (i.e., highway, urban, and grid scenario-based routings) and their limitations. In this review, several recent studies for VANET are selected from urban-, highway-, and grid-based routings. These protocols were selected based on their salient features and grouped together. There are several comprehensive reviews in the literature, but they mainly focus on the components of conventional routing schemes. It is difficult to identify recent developments in QoS optimization by routing protocols in VANETs because parallel research is being conducted in each category of conventional routing and other newly introduced advanced techniques.

This is a novel classification of routing protocols. The goal of this classification is to focus research studies on recently emerged popular schemes and avoid efforts on saturated and obsolete routing schemes. The benefit is to help young researchers analyze current and state-of-the-art proposed algorithms on the conventional side along with newly emerged strategies in routing VANETs. The current state of the art from the last five to seven years is selected to highlight the operation, advantages and disadvantages of the research work, and the specifics of each proposed method. The tables in each subsection show the comparison of the simulations, routing and performance metrics used for the considered routing methods.

## 2. Vehicular Ad Hoc Network Role in Intelligent Transportation System

The vehicular ad hoc network distinguishes itself by its high level of mobility, increased network traffic, and real-time applications, and is essential in the intelligent transportation system (ITS) [7,8]. ITS provides navigation, road safety, traffic control, and electronic tolling services. These applications are primarily real-time and non-delay tolerant. Although there are no battery power constraints in VANET due to onboard units (OBU) installed on vehicles, latency is an issue for its real-time applications.

The rapidly changing topology has therefore drawn researchers to focus on the design of efficient routing protocols. The role of routing protocols in the stated scenario is crucial in supporting the ITS [9]. In the modern era, communication is done through real-time multimedia applications. Establishing a stable link between the nodes is required to ensure efficient and real-time communication, which can be achieved with efficient routing. Efficient routing can control congestion in the network, and a congestion-free network can guarantee the quality of service for real-time applications. The routing protocols must be able to cope with the challenges of dynamic topology and other routing challenges such as quality of service insurance in a rapidly changing scenario.

Furthermore, it must have the flexibility to adjust to the changing requirements. The scope and provision of different routing protocols vary significantly [10]. It is essential to select an appropriate routing protocol for different operational environments automatically.

According to [11], they tested different V2X concepts in a joint project of Michigan University and the National Highway Traffic Safety Administration (NHTSA). Two thousand and five hundred vehicles from well-known automakers such as Toyota, Ford, GM, Nissan, Audi, Mercedes, and Honda took part in the test. NHTSA analysis of the test data shows that V2X can prevent more than half a million road accidents in a year. This technology can save thousands of lives annually.

V2X scope is not limited to collision avoidance; instead, it has several applications that will grow in the future. Figure 2 shows some common uses of this technology. This technology warns the driver about onward collisions and other vehicles' existence at blind intersections. It gives a do not pass warning (DNPW), queue warning, and curve speed warning.

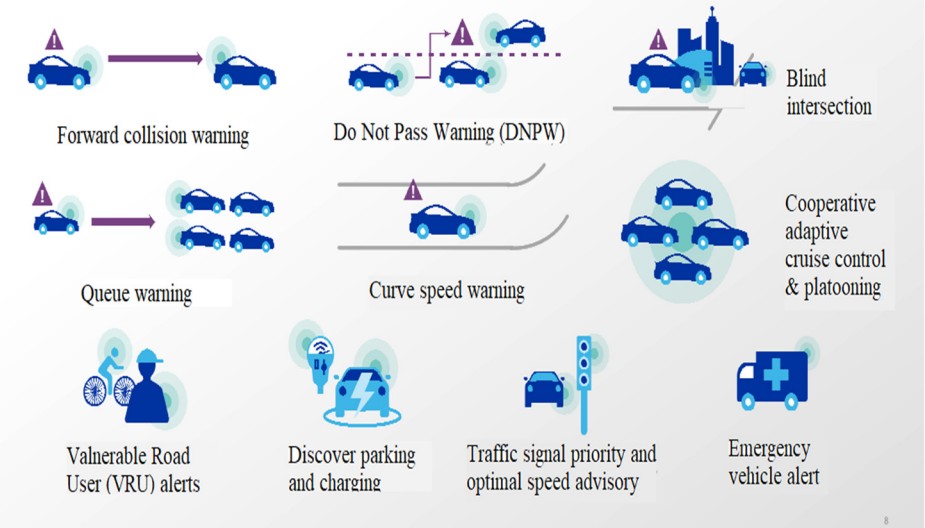

**Figure 2.** V2X technology use cases [6].

This technology discovers parking and charging for the vehicles, informs about traffic single priority, and provides advice about optimal speed. It alerts the driver about vulnerable road users (VRU) and emergency vehicles. The Cooperative Adaptive Cruise Control (CACC) system makes vehicle platooning feasible at small headways [12].

In 2015, MIT Technology declared V2V as a significant technology breakthrough in one of their reviews [11]. Different automakers have announced that they will enable the upcoming models with V2V technology and have shown their interest in capitalizing on the excitement [11]. Delphi Automotive has decided to use NXP's ReadLINK chipset and IEEE 8.11p in V2V modules. To support V2V and V2I applications, Qualcomm has made two X12 and X5 LTE modems and a VIVE QCA65 × 4 chipset [11]. Cohda, Siemens, and GM work together over the radio band for the V2V and V2I devices. Their V2X technology based on a cellular network is called C-V2X [11].

Automaker companies have introduced their initial versions of V2V-enabled vehicles. Still, these vehicles will be able to communicate with only a few others, which is not significant for achieving the real benefits of V2V. According to GM estimation, 25% of vehicles must be enabled with this technology to make it effective [11]. This will require government regulations and a period of about five years [11]. Audi has tested the V2V technology of Delphi, Cohda, and NXP, and Ford also demonstrated vehicles enabled with V2V technology [11].

Toyota is integrating sensors with V2I and V2V technology and safety packages; this technology will be shortly available to the world. Developing technologies such as the Internet of Things (IoT), and products such as traffic lights, signs, crosswalks, various interactive devices, and different wireless products from Savari, Cisco, and Siemens will also be integrated with this technology to benefit from V2V [11].

### 3. VANET Technology

It is necessary to alert the drivers about road conditions, traffic, and other relevant information to ensure traffic flow, safety, and protection. Timely and accurate information is required to achieve this. As illustrated in Figure 3, vehicular ad hoc networks (VANETs) typically solve the issue [13]. By benefiting from the facilities provided by VANET technology, emergencies can be avoided. In other words, all information related to traffic movement on the road, such as traffic density, vehicle speed, and weather conditions, is collected using V2V and V2I communication technologies. This information helps in preventing traffic overflow and road accidents. It also helps the roadside base stations inform the vehicle that the traffic is changing. The V2V network is connected to the external network by integrating various wireless technologies such as 3G, IEEE 802.11, IEEE 802.16e, LTE Advanced, and Long Term Evolution (LTE) [14].

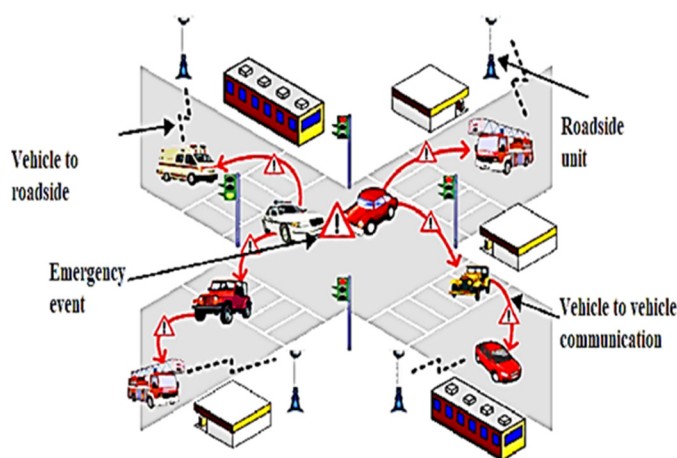

**Figure 3.** Vehicular ad hoc network and ITS.

VANET provides the features of continuous connectivity to drivers while driving; it connects the vehicles with other vehicles through a home or office-based network and enables ITS by establishing stable wireless connectivity to the vehicles without any fixed infrastructure. On-board units update the driver and passenger about floods, accidents, traffic jams, rain, and other traffic irregularities. By receiving in-time information about the road condition, the driver can make the right decision to prevent accidents [9].

The nature of VANET is akin to the operative technology of MANET in that the conditions of auto-organization, auto-management, low bandwidth, and sharing of radio transmission are the same. Nevertheless, the primary operational constraint of VANET is the high speed and timid mobility of mobile nodes along the tracks. This indicates the redesign of the routing protocol, which demands the enhancement of the MANET architecture to perfectly adapt to the high-speed movement of nodes in the VANET. This problem posed several issues in the research for designing an appropriate routing protocol.

The primary purpose of routing protocols is to achieve shorter communication time using as few network resources as possible. Numerous routing protocols are developed for MANETs; some are directly transferable to the VANET. The simulation results show, however, that the efficiency of the VANET is influenced by factors such as high-speed vehicles, active communication, and the resulting high speed of other nodes different from MANETs. Therefore, identifying and administering routes are the required tasks for the VANET. This paved the way for many research issues in developing the appropriate routing algorithm.

A qualitative analysis of protocols shows that geo-casting and position-based routings are better suited than conventional VANET routing protocols due to ambient influences. Position-based routing protocols are based on the geographic position of the vehicles when selecting the best pathway to route the data. Besides, they do not exchange connection status information or maintain fixed routes. This makes the protocols more resilient to frequently changing topologies and vehicles' high mobility [1,15]. Furthermore, infrastructural-based routing protocols are the most attractive in communication with VANET.

*3.1. Manet and Vanet Technologies Comparison*

VANET and MANET are closely related to several technological dimensions. Their variations are also apparent from the characteristics shown in Table 1.

**Table 1.** MANET and VANET comparison.

| Parameters | Manet | Vanet |
|---|---|---|
| Cost of production | Cheap | Expensive |
| Change in network topology | Slow | Frequent and very fast |
| Frequency of topological change | Low | High |
| Bandwidth | Hundred kbps | Thousand kbps |
| Node lifetime | Depends on power resource | Depends on the lifetime of vehicle journey |
| Multi-hop routing | Exist | Less existence |
| Reliability | Medium | High |
| Moving pattern of nodes | Random | Regular |
| Addressing scheme | Attribute-based | Location-based |

### 3.2. Vanet Architectures

VANETs use MANET-like standards because they are not tied to a fixed basis for communications and broadcast messages, and they are part of the highly dynamic road traffic environment. Figure 4 shows the purely cellular/ad hoc, wireless local area network (WLAN) and VANET hybrid architecture. VANETs can use fixed radio access gateways, Wi-Fi access nodes, or base stations at the junction for internet connectivity, routing, or traffic data collection in a strictly cellular architectural environment. Under such circumstances, The VANET will use either Wi-Fi or radio network architecture. Such architecture is known as vehicle-to-infrastructure (V2I) architecture and efficiently incorporates new, disparate wireless technologies, including 3G wireless, LTE, LTE Advanced, IEEE 802.11, and IEEE 802.16e [14].

Figure 4 (red lines) shows the pure ad hoc VANET architecture known as vehicle-to-vehicle (V2V). In this particular structure, nodes are constrained to communicate with one another because financial barriers restrict the deployment of mobile masts and wireless APs. The in-car information collected by the fixed sensors is helpful to inform other vehicles of accidents or other traumas and assist the police in tracking criminals [16]. The infrastructure-free network is located throughout the ad hoc cluster where the nodes carry out V2V communication.

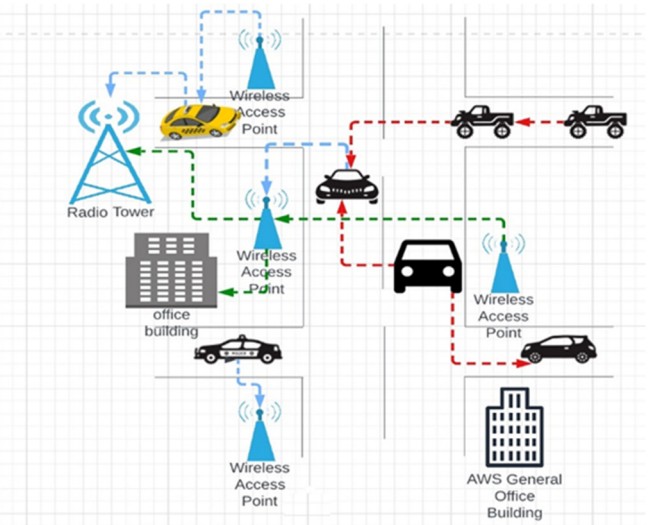

**Figure 4.** VANET architecture [17].

Figure 4 overall presents the (V2V, I2I, and V2I) Hybrid VANET's architecture. This architecture integrates the wireless network components with roadside units like mobile masts, APs, and vehicles to facilitate communication. Many metropolitan screening, safety, driving assistance, and entertainment applications [18] have used infrastructural communicative devices to access live and web-based data across network spaces and communicate such data via ad hoc, infrastructure-free peer-to-peer networks. Hybrid mobile/ad hoc and WLAN architecture deliver more comprehensive content, excellent data-sharing service, and flexibility.

## 4. Routing Protocols in VANETs

In VANETs, routing protocols are commonly divided into vehicle-to-vehicle and vehicle-to-infrastructure based on the architecture of the VANET. Vehicle-to-vehicle VANET is classified further based on routing information and transmission strategies. These are position- or topology-based, unicast, multicast, and broadcast, respectively [19]. The literature also categorized the routing algorithms as reactive, proactive, and hybrid. Predictive mobility-based algorithms and energy-aware routing are other taxonomies. The protocols are designed to ensure the quality of service and efficient use of constrained

resources. Cluster-based routing protocols are also designed to reduce the topology maintenance overhead. Proposed methods are position or topology-based routing maintaining a low-latency, congestion-free network. The recent research in VANETs routing is based on these attributes of routing. Still, their design is broadly based on the scenario where the VANETs will be deployed. In this article, we have categorized the protocols based on the operating scenario, as it will help the researchers focus on studying protocols designed for their intended operational scenario. Figure 5 illustrates the classification of the examined protocols on the basis of their active scenario.

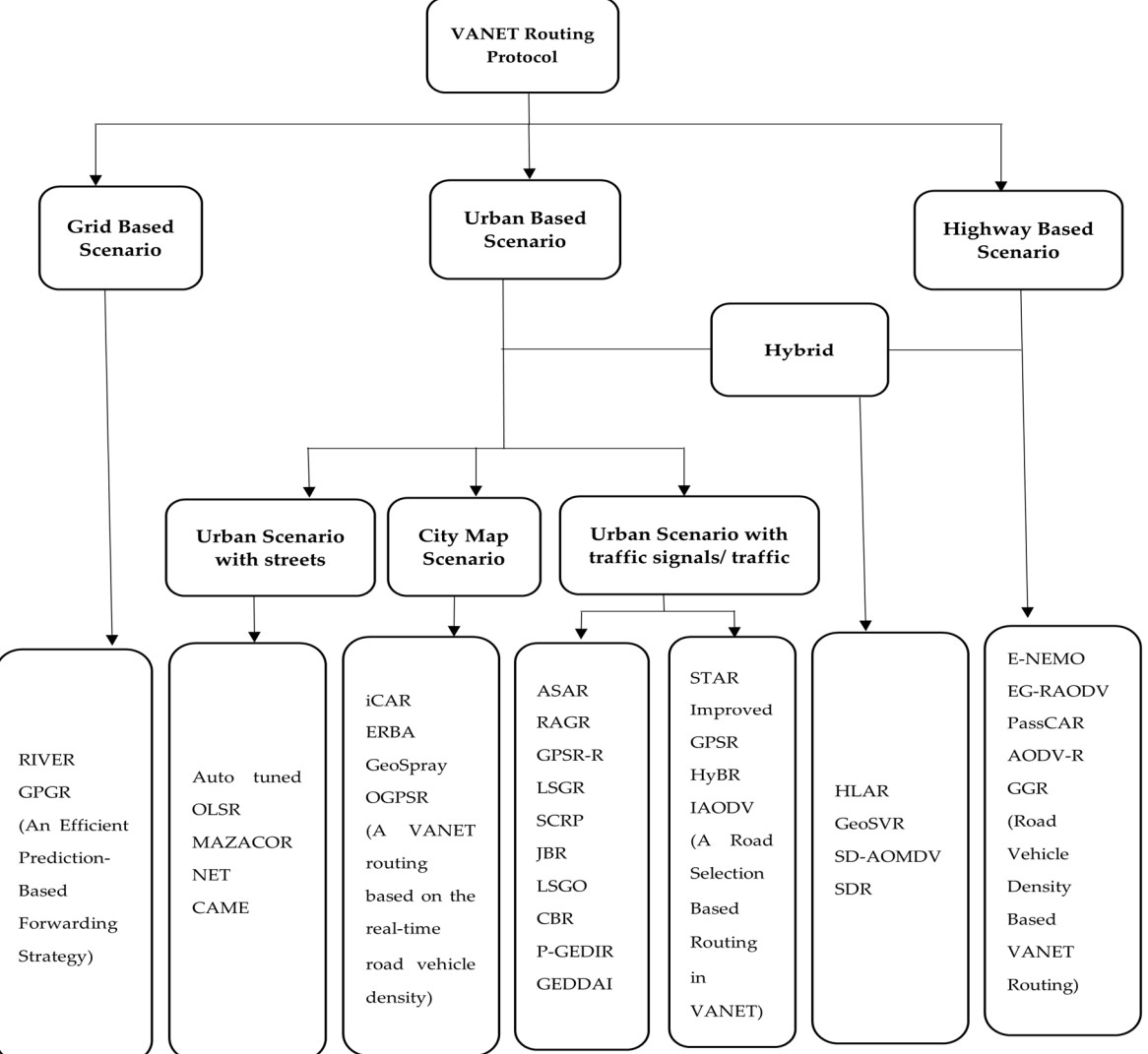

**Figure 5.** Taxonomy of VANETs routing protocols.

There are two reasons why a routing protocol must be developed that can handle topological change in high speed VANETs. First, routing in an ad hoc network is responsible for finding and maintaining a route to the destination, taking into account the characteristics of mobility, limited power, and bandwidth restrictions [20]. Second, VANET dynamics are high, and these protocols have been proposed for particular scenarios.

Continuous research is evolving to improve routing, bearing in mind several aspects and stimulating features of VANETs. This subsection presents these design considerations of routing protocols, the approach adopted in each proposed algorithm, and their limitations. The advantage is to help new scholars to analyze current state-of-the-art proposed systems. Various routing algorithms for VANET selected randomly from different scenar-

ios are described in the sections below. Tables in each subsection demonstrate the parameters used in routing, simulation, and the performance parameters used in the scenario of the routing proposals under consideration.

### 4.1. Highway-Based Scenario

The highway-based scenario represents a multilane road on which the traffic flows at high speed [21]. The highway traffic usually remains sparse, and the internet or inter-vehicle connections are maintained with the help of the roadside unit. The designing aspects of routing protocols intended for highway-based scenarios differ from the city- or junction-based scenarios. Tables 2–4 shows routing parameters, simulation parameters, and performance metrics of highway-based scenarios routing protocols respectively.

### 4.1.1. Enhancement in Network Mobility (NEMO) Protocol

Simple handoffs can be achieved only on IP-based networks if an efficient IP assignment and reassignment process in a mobile node (mn) is implemented. Internet connectivity is provided in vehicle-to-infrastructure communication. On highways, the speed of vehicles is high; thus, seamless handoff and a steady link to the internet are issues.

M-IPV4, M-IPV6, and HM-IPV6 seamlessly process handoffs and are an improved series of advanced routing in IP-based networks. To address network mobility (NEMO) in M-IPv6, the Internet Engineering Task Force (ITEF) RFC (Request for Comments) was expanded and named the NEMO Basic Support Protocol.

An improvement in NEMO to cut down vertical handoff latency was proposed in [22]. The vehicle-to-vehicle communication is employed to invoke the handoff procedure. The nodes/vehicles perform the role of a router for other vehicles during the handoff process along with the fixed infrastructure. The overall latency may be reduced in the pre-handoff process. Wi-Fi and WiMAX wireless networks were used during the simulation process. The proposed scheme was tested for performance metrics such as throughput, packet loss, control overhead, handoff latency, and jitter.

The mobile node requires updated messaging to the home agent to increase the traffic signal ratio. The vehicle-to-vehicle communication has its traffic and is responsible for acting as a router during the pre-handoff process. Hence, the control overhead may increase. Furthermore, the complication arises if the mobile node has two neighbor nodes with different mobility directions. One node is arriving at VANET while another is on departure; what will be the handoff choice of an MN in this situation?

### 4.1.2. Evolving Graph-Reliable Ad Hoc On-Demand Distance Vector (EG-RAODV)

The changes in topology occur when a vehicle/vehicles changes its velocity or lanes. The changes in topology depend on road conditions and drivers' behavior, and are not usually planned. Graph theory is constructive in analyzing the dynamic topology of VANET. For a dynamic network whose mobility is predictable, a theoretical graph model known as an evolving graph can be used to extract the dynamic behavior of mobile nodes [23,24]. The vehicle information and underlying road network can be used to estimate the dynamics of VANET. Therefore, if we assume VANET as a predicted pattern dynamic network, an efficient routing algorithm for VANET can be designed using an evolving graphs model.

Evolving Graph Reliable Ad Hoc On-Demand Distance Vector Routing Protocol (EG-RAODV) based on their former model, known as VANET oriented Evolving Graph (VoEG), was proposed in [25]. To maintain the quality of service in VANET, a procedure based on VoEG creates the utmost trustworthy routes from source to sink nodes.

Evolving graphs are appropriate for the networks whose topology dynamics are predictable at different time interims. The authors assume that VANET topology is predicta-

ble; however, considering that VANET can be characterized under fixed scheduled dynamic networks (FSDNs), this is not acceptable. The EG-RAODV is not compared with the latest routing schemes for performance.

### 4.1.3. Passive Clustering Aided Routing (PassCAR)

The paper [26] proposes a clustering scheme for vehicular ad hoc network based on the passive clustering mechanism adopted in mobile ad hoc networks. The authors claim that the proposed scheme forms reliable and long-life clusters, resulting in decent routing performance. The proposed scheme works in three phases. That is the discovery of routes, route establishment, and data transmission. The selection of suitable candidates for cluster head (CH) and gateway roles is the central theme of this research. The routing information will then be forwarded via these nodes in the route discovery. Multiple metrics are used to evaluate the suitability of a node to become the cluster head. These are node degree, communication workload, and the lifetime of a link. The main factors considered are link stability, sustainability, and reliability. The cars in the network evaluate themselves for the responsibility of the cluster head and gateway role based on the weighted combination of the metrics mentioned above. When the routes are discovered, the source nodes are informed through a route reply packet. The nodes transmit their data on the established paths. The authors claim they are the first to study the passive clustering applicable to vehicle behavior. The association of the proposed scheme to any routing protocol that supports reliable, stable, permanent data delivery and passive clustering-based scheme for the vehicular network that operates on a logical link layer are the main contributions of this paper. A simulation is performed that shows the path's lifetime, throughput, and packet delivery ratio.

In the proposed scheme, the mobility of a car during cluster head (CH) selection is not considered, and a node with a different speed from its neighbors may be selected as a CH, which may result in extra overhead. The neighbor's relative mobility is an essential factor in a structured cluster network and is not considered in this scheme. Therefore, a car with a different direction from its neighbors may be elected as a CH and will form unstable clusters. The degree of a node may be high in a junction and will be selected as a CH, but the CH may become isolated when it leaves the intersection. Therefore, a node with a high degree does not guarantee stable clusters.

### 4.1.4. Road Vehicle Density Based VANET Routing

Geographical routing protocols are more suitable for VANET, as it has features like a global positioning system (GPS) and no limited battery power [27]. The GPSR is a geographical routing protocol based on a greedy approach. It forwards the packets to the neighbors with greed that this passing on will find a path to the destination. When there are no immediate neighbors, the protocol routes the packets to the region's perimeter, and its performance is not good in VANET because of the driver's behavior, changing topology, node speeds, and density. The road layout decides the network topology in a road or highway scenario. The greed of GPSR may lead the packet to the low-density region, and due to connectivity issues, the data delivery may fail.

The paper [28] proposes a routing scheme for VANET based on on-road vehicle density. Considering practical density information, it establishes a route for stable V2V communication in a city environment. The neighbor nodes measure the density from the road information and beacon messages. It ensures a minimum end-to-end delay and provides the best communication route.

The theoretical description of the scheme justifies the delivery ratio improvement, but the improvement in routing overhead and end-to-end delay is not warranted. The road information table maintenance and beacon messages may cause a delay in route establishment, resulting in end to end delay. Secondly, the beacon messaging traffic may

lead to congestion, delay, or even delivery failure. The proposed is checked for performance against a single routing protocol ignoring other known geographical-based routing protocols like GPCR and GPGR.

### 4.1.5. Ad Hoc On-Demand Distance Vector Routing Based on Reliability (AODV-R)

The two most essential issues in VANET routing protocols are scalability and interoperability. Efficient dissemination and routing protocols are required to provide QoS support to various VANET applications. The routing protocols designed for MANET are not suitable for VANETs. In this regard, research work has been done to ensure link reliability. A scheme is proposed in [29] that predicts the link breakage before it happens using the vehicle heading's information. The proposed scheme provides route durability and stability by grouping the vehicles based on their velocity vectors. Velocity aided routing (VAR) [30] is another proposal that selects the forwarding node based on node and destination-relative velocity. It predicts the destination node's future trajectory by analyzing its location and velocity information and forwards the packets to a predicted region. Movement Prediction-based Routing (MOPR), proposed in [31], avoids link breakage and provides table routes by indicating the future position of the node. It uses the vehicle's direction, velocity and location information to predict its future position.

A Reliability-Based Routing Scheme for VANET (AODV-R) proposed in [32] modifies AODV routing protocol with their route reliability definition and link reliability model to provide QoS support in a highway scenario. AODV route establishment is based on the RREQ message broadcasted to the network. The node that receives the RREQ records the previous hop and forwards the RREQ message. When a node finds a route to the destination, it communicates it to the source node through route request-reply (RREP) using the path obtained from previous hop recordings. If a link breakage happens, it is also communicated to the source node through a route error message (RERR). To ensure the link is still active, the AODV sends HELLO messages periodically. The proposed scheme extends the RREQ message with five new fields containing information about node coordinates, speed, direction, and link reliability. The RERR and routing table are also extended with additional field link reliability. The AODV-R uses this information during the route discovery to provide reliable routing.

The rapid change in VANET topology causes route instability. Therefore, the RREQ broadcasting in VANET is more frequent; this causes high congestion in the network and leads to high end-to-end delay. The proposed scheme increases the size of the RREQ message, increasing the bandwidth load. The increased computational overhead due to link reliability calculation may also cause a delay. The proposed scheme needs to be checked for end-to-end delay.

### 4.1.6. Ad Hoc On-Demand Distance Vector Routing Based on Reliability (AODV-R)

Dedicated short-range communication (DSRC) is designed to enhance the Wi-Fi technology for the VANETs environment. The distinguishing feature of DSRC is the high data rate in a rapidly changing environment. The rapidly changing environment causes critical issues such as data dissemination or efficient routing in ITS applications. The epidemic routing (ER), probabilistic routing protocol using the history of encounters and transitivity (PROPHET), spray and wait (S&W), and DTN to VDTN are different protocols proposed for highway scenarios. These protocols are based on an enhanced version of flooding. These protocols lack the provision of NHV selection. These protocols suffer from lower packet delivery ratio, packet loss, higher hop-to-hop disconnection, end–end delay, low throughput, and high hop-to-hop count.

Kaiwartya, O. and S. Kumar in [33], using a guaranteed geocast routing (GGR) protocol, had proposed guaranteeing the packet delivery in intermittently connected highway VANETs. The proposed protocol considers caching of packets, neighboring vehicle speed, packet ownership transfer, and heuristic function with NHV selection. Cached packets are not immediately forwarded due to the unavailability of NHV in intermittent

connectivity. The mobility is used to deliver the cached packets upon NHV availability. A mathematical model computes the probability of the successful delivery of packets in the intermittently connected highway without considering caching packets. It divides the vehicles into the groups FAST and SLOW based on their speed compared to a current forwarder. The FAST and SLOW impact on end-to-end delay is analyzed. The packet delivery is guaranteed through ownership transfer. The present and future costs of packet delivery are measured by the heuristic function that helps in NHV selection.

Several parameters used in the proposed protocol and mathematical models may increase the processing and communication overhead significantly. The simulation environment of the proposed scheme does not reflect the real-world environment. The speed range is 40–120 km/h in a real-world highway scenario, which is kept at 50–60 km/h in the simulation setting of the proposed scheme. The impact of speed is very high on routing protocol performance in VANET.

**Table 2.** Routing parameters of highway scenario-based routing protocols.

| Article | Name of the Proposed Protocol | Year of Proposal | Routing Parameters | | | | | |
|---------|-------------------------------|------------------|--------------------|--|--|--|--|--|
| | | | MAC Protocol | Transmission Range | Operational Scenarios | Speed | No. of Nodes | Topology Size |
| [22] | EG-RAODV | 2013 | NA | NA | Highway with three lanes | 40,60 and 80 km/h | 30 | 5000 m highway |
| [23] | AODV-R | 2012 | NA | NA | Highway scenario with three lanes | 40–80, 60–140 and 40–100 km/h | 30 | 5000 m with three lanes |
| [24] | GGR | 2015 | IEEE 802.11p | 300 m | Highway scenario with 6 lanes | 50–60 km/h | 10–50 | 50 km * 50 km |
| [25] | Road Vehicle Density-Based VANET Routing | 2013 | IEEE 802.11 | 250 m | Highway | 20–55 km/h | 150–300 | 3000 m * 3000 m |
| [26] | PassCAR | 2012 | IEEE 802.11 | 250 m | One way multi-lane platoon scenario | 80,100, and 120 km/h | 150, 200, 250, 300 and 350 | Road length = 5 km |
| [27] | Enhancement in NEMO | 2013 | NA | WiMAX = 1000 m, WLAN = 300 m | Highway with four lanes | 5–100 km/h | 0–100 | 1000 m × 1000 m |

**Table 3.** Simulation parameters of highway based scenario routing protocols.

| Referenced Article | Simulation Parameters/Metrics | | | | | | | |
|--------------------|-------------------------------|--|--|--|--|--|--|--|
| | Simulation Tool | Compared to | Packet Size (Bytes) | Data Rate (kb/s) | Traffic Type | Channel Capacity | Simulation Time | Mobility Models |
| [22] | OMNet++ | AODV, PBR | 1500 | 128 | NA | NA | NA | NA |
| [27] | Ns-2 | NEMO, fast NEMO | 320 | 100 packet/s | NA | NA | NA | NA |
| [26] | Ns-2 | AODV | 1000 | NA | NA | 1 mb/s | 100 sec | One way multi-lane platoon scenario |
| [25] | Ns-2 | GPSR | NA | 20–40 | NA | NA | 200 sec | Highway |

| [24] | NS-2.34 and MOVE | ACSF, S&W and ER | 512 | NA | NA | NA | NA | Highway scenario |
| [23] | OMNET ++ and C++ | AODV | 500–3000 | 10 packets/s | UDP | NA | NA | Highway scenario with three lanes |

**Table 4.** Performance metrics of highway scenario-based routing protocols.

| | Performance Metrics | | | | | |
|---|---|---|---|---|---|---|
| Reference | Packet Delivery Ratio | End to End Delay/Average Delay | Throughput | Packet Loss | Routing/Message/Communication Overhead | Other Metrics |
| [22] | Yes | Yes | No | No | No | Link failure |
| [27] | No | No | Yes | Yes | No | Message overhead |
| [26] | Yes | No | Yes | No | No | Path lifetime |
| [25] | Yes | No | No | No | Yes | NA |
| [24] | Yes | Yes | Yes | Yes | No | Hop to hop disconnection |
| [23] | Yes | No | No | No | No | Routing error message |

### 4.2. *Hybrid of Highway and Urban/City Scenario*

In this section, those protocols are discussed and designed for a scenario with both the highway and urban scenarios. The urban scenario is restricted in nature due to streets and buildings. The speed of vehicles in the city is low as compared to highways. The traffic remains congested in the city, and the connectivity is high due to the maximum number of access points in city areas. Tables 5–7 shows routing parameters, simulation parameters, and performance metrics of hybrid of highway and urban based scenarios routing protocols respectively.

### 4.2.1. Hybrid Location-Based Ad Hoc Routing (HLAR)

In VANETs, issues like wireless channel fading, vehicle scalability, and density variations have made routing more challenging. These issues may arise due to high-speed vehicles and common obstacles in urban sceneries. An adequate volume of research work focuses on realistic mobility and propagation models [34–39]. These proposals attract significant concerns that a new VANET routing protocol may face. VANET routing schemes are categorized primarily into geographic or position-based and topology-based routing [40–43]. The link-state information is used to data forward data in topology based routing schemes. The routing overhead in ad hoc on-demand distance vector (AODV) topology-based schemes is low compared to others [34,44–46]. The scalability issue arises in all topology-based routing schemes [45]. Topographical-based routing schemes do not share information about their links as they are not table-driven. This overcomes the scalability issue to some extent.

A hybrid location-based ad hoc routing (HLAR) scheme to handle the scalability issue was proposed in [47]. The commendable characteristics of topographical and reactive topology-based routing are combined in this protocol. The reactive version of HLAR is activated when topographical information is not available. The simulations are conducted to evaluate the performance of network overhead and scalability.

Real-time communication is mandatory for most VANET applications. The rout formation in the initial phase may cause a delay in reactive routing. The scalability and routing overhead is checked during simulation, while the packet loss ratio and latency parameters are ignored in the simulation. The mean speed of vehicles is kept constant for diverse scenarios where it would differ since the speed of vehicles in streets, and multi-line highways may vary. This may result in non-realistic outcomes during simulations.

### 4.2.2. Speed and Direction Based Ad Hoc On-Demand Multipath Distance Vector (SD-AOMDV)

In MANET routing protocols, AODV is the most suitable routing protocol for VANET. Marina and Su in [48,49] verify its performance in terms of packet delivery ratio, path optimality, and routing overhead by simulating it against other ad hoc routing protocols such as DSDV, TORA, and DSR. An improvement in AODV is proposed in [50] that uses mobility parameters for next-hop selection and makes it adaptable for VANET. In this scheme, the selection of a node as the next hop is based on its direction and position. AOMDV is another enhanced version of AODV, which is better than AODV for VANET's high mobility scenarios [51].

Further improvements have been added to AOMDV to make it suitable for V2V communication. In S-AOMDV [52], routing decisions are based on hop and speed. Its performance is better than AOMDV in terms of end-to-end delay and normalized routing load. RAOMDV [53] considers the number of hops, link quality, number of retransmission, and delays as parameters to enhance routing. It is a multipath routing scheme that reduces the number of route rediscoveries.

The proposed SD-AOMDV [54] is based on AOMDV that enhances AOMDV suitability for VANET. The design considerations are based on the fact that high-speed difference and different direction of the communicating nodes reduces the route and link stability. In this scheme, the next-hop selection is based on the node, source, and destination similarity concerning speed and direction. It selects only that intermediate node as the next hop in the source and destination path. If the source and destination nodes are in the opposite direction, then it selects only that intermediate node as the next hop in the direction of the source or destination. The intermediate node is also checked for its minimum speed difference with source and destination before selecting the next hop. The simulation verifies the performance of the proposed scheme.

In the city scenario, the direction of the nodes changes frequently, and a node selected in the direction of source and destination may change its direction. If a node is at a red light signal at the junction, its direction is uncertain because of the unknown driver's intentions. S-AOMDV and RAOMDV are the other improved AOMDV routing schemes that claim improved performance. The proposed scheme simulation validates its better performance than AOMDV, but its performance against S-AOMDV and RAOMDV is not verified.

### 4.2.3. Stable Direction-Based Routing (SDR)

Topology-based routing use link information for route establishment. In routing discovery, route request (RREQ) packets are broadcasted to the neighbor nodes. The neighbor nodes rebroadcast the packet, and this rebroadcasting continues until route establishment. This is an effective solution to rout discovery, but not efficient, as it broadcasts the packet throughout the network. Position-based routing protocols send packets to a specific region based on a known node position. The position of communicating nodes and intermediate nodes is known, and the packets are forwarded towards the destination without establishing a route. Position-based routing is comparatively good for VANET, but these routing protocols suffer from link failure during broadcasting. Link stability is required to guarantee minimum end-to-end delay.

Liu, C. et al. in [55] proposed a stable direction based routing (SDR) that broadcasts the RREQ packets in a specific direction based on destinations and neighbor node position. It minimizes the flooding and its negative effect on the network. It also provides stable links for propagation by measuring link stability and labeling paths with path stability and predicted expiry time. It considers the vehicle in the direction of source or destination in the route discovery process. The protocol predicts the path duration and selects stable links as a route. The SDR reduces the flooding by using directional broadcasting of RREQ and reduces network delay by selecting a stable route.

SDR is a hybrid of topological and geography-based routing protocol; it reduces the congestion problem of topology-based routing and links instability problem of geographical routing on the cost of computational overhead. The calculation and maintenance of link stability are continuous and can cause network delay. Secondly, RREQ broadcasting causes congestion, although it is reduced compared to topology-based routings.

4.2.4. Geographic Stateless VANET Routing (GeoSVR)

In an urban scenario with multi-hop forwarding, finding a suitable route for the data packets is an issue. Dynamic topology, high speed, and non-uniform density are the natural characteristics of VANET that cause the issue mentioned above. Secondly, wireless communication is a short-distance communication that can be easily interfered by different obstacles. This obstructs routing significantly. Traditional routing schemes cannot route packets with high packet delivery ratio and low latency, which are the essential routing parameters for urgent messaging. Reactive routing such as AODV and DSR are unsuitable for VANET as they do not satisfy the minimum latency requirement of traffic management applications.

The potential solution to the problem is geographic routing. It reduces the latency and overhead as the geographic routing does not require exchanging route maintenance information or link status. GPSR is a typical routing protocol of this category; however, its scope does not cover the roadways scenarios with variable node density and sparse connectivity. GPCR [56] solves the problem of GPSR, but fails to provide a complete solution for the local maximum problem. GSR is another solution, but the sparse connectivity problem is not covered here.

In [57], Geographic Stateless VANET Routing (GeoSVR) based on optimal forward-path (OPF) is proposed. A Restricted Forwarding Algorithm (RFA) is used to overcome sparse connectivity, unreliable wireless communication problems, and local maximum. It is shown in the simulation results that the proposed protocol outperforms other routing protocols.

The authors have described the unsuitability of AODV reactive routing and suggest geographic routing for VANETs, but simulate their own proposed scheme against AODV instead of the GPCR. The node density is not apparent, as the simulation area parameter is missing in simulation parameters. The proposed protocol is checked for only two performance metrics.

**Table 5.** Routing parameters of hybrid scenario-based routing protocols.

| Article | Name of the Proposed Protocol | Year of Proposal | Routing Parameters | | | | | |
|---------|------------------------------|------------------|--------------------|--------------------|---------------------|-------|----------------|----------------|
| | | | MAC Protocol | Transmission Range | Operational Scenarios | Speed | No. of Nodes | Topology Size |
| [28] | (HLAR) | 2012 | IEEE 802.11b | 150–250 m | Gaussian, Rayleigh, Uniform | 40–100 km/h | 40–240 | NA |
| [29] | (GeoSVR) | 2012 | IEEE 802.11 | 250, 600 m | Line, urban, static and mobile scenario | 20–80 km/h | 1000, 150, and 3 (for static scenario) | NA |

| [30] | SD-AOMDV | 2012 | IEEE 802.11 | 250 m | City scenario, highway scenario | 10–90 km/h, 60–120 km/h | 70, 60 | 2000 m * 2000 m |
| [31] | (SDR) | 2013 | IEEE 802.11 DCF | 400 m | City scenario with intersections, highway scenario with three lanes | 10–35 m/s | 60–600 | 36 km * 50 m |

**Table 6.** Simulation parameters of hybrid scenario-based routing protocols.

| Referenced Article | Simulation Parameters/Metrics | | | | | | |
| | simulation Tool | Compared to | Packet Size (Bytes) | Data Rate (kb/s) | Traffic Type | Channel Capacity | Simulation Time | Mobility Models |
|---|---|---|---|---|---|---|---|---|
| [29] | Ns2 | AODV and GPSR | 800 | 400 kb/s | NA | 2 mb/s | NA | Line, urban, static and mobile scenario |
| [28] | NA | AODV-ETX, MTL | NA | 8 kb/s | NA | 2 mb/s | NA | Gaussian |
| [30] | NS-2.34 | AOMDV | 512 | NA | NA | NA | 400 sec | Manhattan mobility model |
| [31] | Qualnet 4.0 | AODV, D-LAR, EARP and ROMSGP | 1024 | NA | NA | 2 mb/s | 900 sec | City/highway scenario |

**Table 7.** Performance metrics of hybrid scenario-based routing protocols.

| Reference | Performance Metrics | | | | | |
| | Packet Delivery Ratio | End to end Delay/Average Delay | Throughput | Packet Loss | Routing/Message/Communication Overhead | Other Metrics |
|---|---|---|---|---|---|---|
| [29] | Yes | No | No | No | No | Network latency |
| [28] | Yes | Yes | No | No | Yes | NA |
| [30] | Yes | Yes | No | No | No | Normalized routing load |
| [31] | Yes | Yes | No | No | No | Path breakage |

### 4.3. Urban Based Scenario

The limited speed of vehicles in urban areas causes congested traffic. The design considerations of routing protocols in urban areas have distinguished dimensions [58]. The urban-based scenario is further divided into an urban scenario with traffic signals/traffic, a city map scenario, and an urban scenario with streets. Tables 8–10 shows routing parameters, simulation parameters, and performance metrics of urban-based scenarios routing protocols respectively.

### 4.3.1. Urban Scenario with Traffic Signals/Traffic

This scenario represents an urban scenario that considers the junctions' traffic signals. The behavior expected direction of the stopped vehicle at the intersection and the congestion at traffic signals make the scenario different from highway and city map scenarios.

(1) Peripheral Node-Based Geographic Distance Routing (P-GEDIR)

Routing is finding the best path between the source and destination. Source and destination may contain multiple hops in between; this situation is more complicated than a one-hop communication. The intermediate vehicles act as a router in determining the traffic path. Frequently changing network topology in VANET makes it very hard to find and maintain the routes. Position-based routing protocols are more suitable for VANET than the traditional topology-based routing protocol. GPSR, A-Star, GREEDY PERIMETER COORDINATOR ROUTING (GPCR), MFR, and GEDIR are the known position-based routing protocols.

In [59], they analyze the performance of a location-based routing protocol, Peripheral node-based GEographic DIstance Routing (P-GEDIR), based on the GEographic DIstance Routing (GEDIR) protocol. P-GEDIR reduces the number of hops in the route, improving data delivery in the urban traffic scenario. The number of hops between source and destination is reduced using the concept of the peripheral node.

The author claims that the analyzed scheme improves the packet delivery in various VANET scenarios, but it is not validated in the simulation. The result does not show that overall QoS performance is enhanced with the scheme implementation. The proposed scheme is not checked against variable speed and node distance.

(2) Geographical Data Dissemination for Alert Information (GEDDAI)

One of the most challenging and essential processes in VANET is data dissemination. VANET natural features such as frequently changing topology, disconnectivity, and variable node density make data dissemination challenging. The efficient and robust data dissemination is necessary for accident avoidance and after collision warning, particularly when the source and the destination distance exceed their radio transmission range. Issues such as broadcast storm, network partition, and temporal network fragmentation must be resolved efficiently to achieve efficient and robust data dissemination for VANET.

The paper [60] proposes geographical data dissemination for alert information (GEDDAI) that efficiently solves the broadcast storm problem. It reduces the delays and overhead by performing data dissemination across the relevant zones utilizing proposed sweet spots. The designed protocol is based on a reactive approach, avoiding the table-driven technique, which is very costly in VANET due to its frequently changing topology.

The zone maintenance, management, and formation will cause additional overhead. The proposed protocol is close to the cluster-based scheme as it divides the operational environment into zones, and its performance is also supposed to be checked against cluster-based schemes. Unlike the sweet spot, the zone of relevance (ZoR) decision shown in the algorithm flowchart is not clearly described.

(3) Shortest-Path-Based Traffic-Light-Aware Routing (STAR)

Multi-hop relaying among nodes is used to achieve packet forwarding in VANETs. Features like frequent changes in topology and speed are the reasons due to which end to end connectivity is not ensured in VANETs. VANETs have constrained mobility due to speed limits, obstacles, and roads. The routing and forwarding schemes designed for various situations (e.g., roadways, rural, or urban) may not be the same because of different requirements. Numerous new routing protocols are designed to handle these issues. Greedy forwarding, along with carry and forward, is one of the promising routing strategies designed to solve the frequent disconnection issue in VANETs packet forwarding.

In this regard, the literature has proposed intersection-based routing protocols with traffic lights considerations. The scenario for such schemes is an urban area with high node density in which the nodes/car mobility pattern is stop-and-go. The carry and forward, besides the greedy forwarding mechanism, is used to deliver packets to the destination nodes moving in between intersections. The decision of forwarding at an intersection is either in a straight direction or diverted towards steep roads. The decision depends on the destination location and road vehicle distribution. Here, the issue is the green and red lights that control the traffic flow and consequently affect the VANET end-to-end connectivity.

The paper [61] tackles the problem with Shortest-Path-Based Traffic Light Aware Routing (STAR), a novel intersection-based routing protocol for an urban area VANET. The Green-Light-First (GLF) scheme does not ensure efficient performance. Red lights at intersections increase vehicle density. The proposed scheme analyzes the gathered vehicles for link connectivity probability. The proposed scheme performance is evaluated in terms of packet delivery ratio and network latency against GyTAR, VVR, and GLF using the ns-2 simulator.

The scenario under consideration is defined with the author's assumptions that have missed some realistic traffic flow features. The direction of vehicles at the junction of green and red lights is ignored. The density on red lights is high, but what is the probability that the proposed scheme will always choose the nodes in the direction of the destination for packet forwarding? The author rejects the GLF due to its occasional performance and develops a scheme that is based on probability.

(4)　Improved Geographic Perimeter Stateless Routing

The studies on VANET routing performance show that the position-based routing strategy GPSR is more suitable for VANET routing as the simulation results show its better performance in terms of packet delivery ratio and delay. Hence, many improvements and variations in GPCR are proposed, such as GSR deploying GPSR in the city environment. The Dijkstra algorithm identifies the shortest path between source and destination on a digital map. GPCR is also based on GPSR with a modification in packet forwarding strategy. GPCR does not forward the packets to the streets across junctions; instead, it uses a greedy algorithm and forwards packets to the junction nodes. The Geographic Perimeter Stateless Routing Junction+ (GPSRJ+) [62] is another strategy based on GPSR that modifies the perimeter mode to reduce the packet load at junctions. Brahmi et al. in [63] propose a lifetime concept to minimize the effect of vehicle speeds on GPSR.

The strategy proposed in [64] suggests Hello Packet with the vehicle moving direction, speed, density, and priority flag for adequate route assurance. Through Hello Packets, the vehicle is informed about the neighbor's location and neighbor future location. The node is selected for forwarding packets based on one-hop neighbor priority. The GPSR is designed for a generic ideal scenario and may suffer from local maxima. The proposed strategy recovers the routes by buffering the preliminary data and forward route recalculation.

The proposed strategy does not consider the GPSR message delay. Some modified and improved versions of GPSR, such as GSR, GPCR, and GPSRJ+. The proposed strategy was supposed to be checked against these improved strategies and GPSR. The Hello packet will require extra bandwidth utilization, and due to Hello Packet traffic, the congestion may occur that will result in message delay.

(5)　A Hybrid Bio-Inspired Bee Swarm Routing Protocol (HyBR)

Designing an efficient routing protocol is a challenging task. The passengers need real-time information from road safety services to make safe decisions. The two most crucial requirements for this is maximum packet delivery ratio and end-to-end delay. In sparse networks, when the source and destination are out of their respective radio transmission range, V2V and V2I communication cannot satisfy the constraints of road safety applications.

Hybrid Bee swarm Routing (HyBR) [65] is a unicast routing protocol proposed for VANETs. It uses topology-based routing for the dense network and geography-based routing for the low, dense networks inspired by the bees' communication and bees' marriage, respectively. It's a multipath routing protocol guaranteeing the VANET road safety application requirements. The source initiates route request packets known as forwarding scouts and sends these packets to its neighbors. The forward scouts move forward as the same process is repeated until it finds the destination or until a route to the destination is discovered. When the route to the destination is discovered, a route reply known as a

backward scout is generated and is dispersed to the source. If the forward scout is encountered with multipath discovery, the proposed strategy uses a genetic algorithm (GA) to select an optimal route based on the geographic coordinates of the network. The proposed approach is simulated for a realistic mobility model for end-end delay and packet delivery ratio against AODV and GPSR routing protocols.

The proposed routing strategy selects an optimal route to the destination using GA, which suffers from early conversion that may lead to non-optimal route selection. Secondly, GA requires heavy processing and is not suitable for real-time applications. The proposed strategy is not compared to the improved versions of AODV and GPSR.

(6) Improved Ad Hoc on-Demand Distance Vector (IAODV)

An efficient routing protocol for VANET is reliable, robust, and has minimum latency and network load. In topology and position-based routing, the routing protocol forwards the packets to the destination by using the intermediate nodes as a relay. Among other topology-based routings, AODV is efficient in normalized routing load and packet delivery ratio. Still, its performance is low in packet delivery ratio and end-to-end delay. On the other hand, AOMDV is efficient in minimizing packet drops, and DSR efficiently reduces end-to-end delay. AODV can be a better routing choice in VANET if optimized for the end-to-end delay and normalized network load.

An improved AODV (IAODV) is proposed in [66] to enhance overall routing performance by combing the efficient features of AODV, DSR, and AOMDV. It provides a high packet delivery ratio with minimum end-to-end delay. Further, it provides a route with a minimum number of hops along with a backup route to the destination. The proposed routing scheme is designed to modify the route request as a limited source of up to two hops and reply for a backup routing procedure. In case of broken links or route failure on the primary route, the packets are transmitted to the destination using the backup route. It rediscovers the route if the backup route also fails in packet delivery. The overall working mechanism can be divided into two phases, route discovery, and maintenance. The authors simulate the proposed scheme for performance under a realistic city scenario using NS-2 as a simulation tool.

The simulated scenario would be more realistic if varying vehicle density, varying active connections, and variable vehicle mobility were accommodated into a single scenario. The proposed routing scheme is a hybrid of AODV, DSR, and AOMDV. Its performance is supposed to be checked against their performance, whereas it is simulated against AODV only.

(7) Adaptive State Aware Routing (ASAR)

Position-based routing protocol GPSR forward packets based on geographical location using a greedy algorithm. It reduces the topology change's effect, but suffers transmission delay when the packets are sent to the sparse or low-density region. The GSR uses a city map and discovers the shortest path to the destination using the Dijkstra algorithm. It considers the junctions but not the connectivity resulting in packet loss. A-STAR considers a region with bus routes as the density will be high on those routes. It labels every section with weight and, using the Dijkstra algorithm, finds the shortest route to the destination. A-STAR's procedure to label sections with weight is static.

The issues described above are attempted to be addressed in [67], which proposes Adaptive State Aware Routing (ASAR). The proposed scheme provides a high data rate with low end-to-end delay and is free of the topology change effect. It collects the traffic information from the roadside units at junctions. The roadside units use the transmission delay model based on density to calculate the expected transmission delay. ASAR forward data through the fixed road equipment are on a path that is determined as a low transmission delay path by the fixed units. The proposed scheme is simulated for the performance evaluation against GSR and GPSR in terms of packet delivery ratio and end-to-end delay.

The proposed scheme is based on the desired scenario with equally distanced junctions throughout the city. The scheme is based on fixed equipment at junctions, but if the fixed equipment is out of the source node's transmission range, the route establishment policy of the proposed scheme may not work. The scheme must be checked for routing overhead as a model is used to estimate the route's transmission delay.

(8) A Road Selection Based Routing in VANET

Dedicated Short Range Communication (DSRC) for VANET provides support for V2V, V2I, and Infrastructure to Vehicle (I2V) communication to make ITS service possible [68,69]. High mobility in VANETs causes disconnection in communicating vehicles, resulting in a disruption in ITS service. Network gaps affect the communication system's performance due to increased delay in data transmission. Topology-based routing protocols are not suitable for VANETs due to dynamic topology. Position-based routing schemes are ideal for VANETs where path maintenance is not required. The data transmission to the destination in position-based routings is based on the position information.

Road selection-based routing, proposed in [70], predicts the network gaps in the route. The proposed scheme is a novel scheme for data transmission without delay. Every road in an operational environment is rated with expected delay in data transmission between junctions, shortest paths, and average speed of the vehicles. A static controller node at the intersection is used to calculate the ratings for connected roads. The proposed scheme proposes a path recovery procedure to cover the link breakage problem caused by network gaps.

The proposed routing scheme rates the connected roads to the junction with the shortest path, expected transmission delay, and average speed. If the destination node is on a highway rated with high delay and long path, the data is supposed to be transmitted in the direction of the destination regardless of the proposed scheme ratings. A load of overall communication will be converged to the static node, which may cause high delay and high routing overhead. The static node will gather the information to calculate road ratings, which is an additional communication and computational overhead.

(9) Road Aware Routing Protocol (RAGR)

The geographical routing protocols possess multiple merits over topology based routing protocols as they forward data toward long-distance destinations with significant progress. Geographic routing protocols, however, have difficulty in identifying an optimal path and picking the next most suitable hop because of the volatile nature of the links in the urban scenario, intermittent connections, and signal attenuation. To overcome these issues, it is necessary to design a routing protocol that considers the appropriate and adequate metrics like distance, traffic density, and distance for forwarding data in the multi-hop urban scenario and high mobilities in the VANET.

In [71], Road Aware Routing Protocol (RAGR) is proposed for forwarding data packets in urban areas. Using distance, traffic congestion, and directional routing metrics, the proposed protocol is designed to solve the packet loss and delay problems in urban VANETs. RAGR uses distance and directional information to select the best node for forwarding data in the network. It selects the next route at junctions on the basis of connection quality, destination distance, and analysis of the vehicle density. The performance of the proposed protocol is tested against GyTAR, SDR, and CGMR, using NS-2 simulator.

There are two processes in the proposed protocol: next forwarding node selection and next route selection at the junction. The two operations require computation and a set of information that can increase the overhead of routing. Maintaining the required information requires additional communication.

(10) Stable Connected Dominating Set-Based Routing Protocol (SCRP)

The network environment is necessary for infotainment applications achieving higher throughput and avoiding transmission delay in ad hoc vehicular network. This is not easy to achieve in a city scenario, as estimating the density of vehicles in a region is

difficult because of variations in traffic flow between day and night and between downtown and suburban areas. The distribution of vehicles across different regions is uneven as the density of vehicles converges at intersections. These challenges and obstacles in an urban scenario make intersections ideal regions for making route decisions. A set of routing protocols is proposed to address these observations in a greedy approach. In GPCR, GPSR, and GSR, the routing decisions are based on the shortest path between the source and the destinations. In RBVT, A-STAR, GyTAR [72], and IGRP [73], they select well-connected road sections for forwarding packets to the destination. They suffer from the congestion and local maximum problem because of the greedy approach.

Proposed in [74], the stable CDS-based routing protocol (SCRP) is a distributed geographic routing technique. The SCRP bases routing on a global network topology selecting routes with minimal end-to-end delay. It computes end-to-end delay for a route prior to the data being transmitted. In SCRP, the vehicle speed and spatial distribution are taken into account to develop backbones on road segments using the Connected Dominating Set (CDS). At intersections, a bridge node links the backbones and tracks delay using updated network topology. SCRP uses such information and assigns a weight to each road segment. It creates a route using low-weight road segments.

In the SCRP, no predefined mechanism is used for backbone maintenance. In a flat network, scalability problems may be encountered due to the lack of routers and mobile vehicles in VANETs. The local maxima issue of the greedy scheme is eliminated at the expense of routing and computational overhead.

(11) Junction-Based Geographic Routing (JBR)

The topologies in VANETs are not totally random, although they are dynamic. The node's movement in VANETs is predictable as the movement is restricted to the layout of the roads. This predictability is good for improving link selection, but the number of paths to the destination decreases due to linear topology. VANETs are scalable networks, and in an urban environment, the obstacles, junctions, and traffic jams cause bandwidth issues. The success of VANETs lies in an appropriate routing protocol. The geographically based routings are accepted as predominant as the restricted movement of nodes can be predicted using street maps, navigation systems, and traffic models.

A geographical-based routing protocol is proposed in [75] that uses a greedy approach to deliver data to the destination without delay. The proposed scheme forwards the data packets towards the destination by the junction to junction forwarding strategy; therefore, it is called Junction Based Routing (JBR). It invokes a recovery model when a local maximum issue arises. The recovery model provides a safe and accurate solution to the problem. JBR determines the next best hop selection using the minimum angle method.

It is an additional overhead to detect the local optimum problem and then call another model for its recovery. There are many proposed improved versions of AODV and GPCR; the proposed scheme performance needs to be checked against these improved versions and the original GPCR. The street's intersection and road junctions are not the same as they have different node densities and distances between two consecutive intersections/junctions.

(12) Link State Aware Geographic Opportunistic Routing (LSGO)

The greedy forwarding strategy in geography-based routing makes hop transmission closest to the destination. However, it faces the issue of link reliability due to the transmission range limitation of the communicating end nodes and their mobility. A forwarding strategy is proposed in an opportunistic routing that utilizes the broadcast characteristic and provides backup links for data transmission to improve the link's reliability. It increases the opportunities for the packet to be received. The opportunistic routing schemes have variations in routing metrics considerations; the hop count, distance to the

destination, energy, and cost are the different routing metrics that have been given preference in various schemes. Some of them combine geographical location with link-state information.

A link-state aware Geographic Opportunistic routing protocol (LSGO) is proposed in [76] with a forwarding strategy based on link-state information and geographic location. A mechanism is used to develop a set of candidate nodes. The candidate set is a list of forwarders selected based on the link's quality and geographic location. The enhanced ETX metric measures the link quality. ETX metric shows the expected number of transmissions to choose the next hop. A timer-based scheduling method is used to prioritize the forwarders. The proposed routing protocol can perform very well regarding the packet delivery ratio and reliability of transmission links.

To provide backup links, multicasting to a group of neighbors is needed. This will increase network routing overhead and usage of network resources. The link quality may change over time due to variations in operational environments. The proposed scheme needs to be checked for performance validation under different environmental scenarios.

(13) Link Reliability Based Greedy Perimeter Stateless Routing (GPSR-R)

GPSR studies carried out in [77] state that in VANETs, the nodes frequently reposition themselves and may not be able to provide updated position information to the source node; this may lead to wrong forwarding decisions. When the greedy forwarding fails, the GPSR forward packets to the destination node using perimeter forwarding mode. The perimeter forwarding causes an increase in end-to-end delay as it encounters a high number of hops to reach the destination.

The authors in [78] present a reliability-based GPSR protocol (GPSR-R). The proposed scheme is designed for the highway scenario. It checks the reliability of a communication link by using the link reliability metric before selecting the one-hop forwarding vehicle. It measures link reliability using an analytical model. The analytical model defines the link duration probability by using the nodes' direction and speed. The simulation results show that the proposed scheme outperforms the conventional GPSR and provides high throughput and packet delivery ratio.

The nodes are supposed to maintain a list of neighbors that will be updated periodically with beacons, which may cause communication overhead. The analytical model computes link reliability for a communication link, which may cause computational overhead and delay. The proposed scheme is validated under a specific scenario where all the vehicles are moving in the same direction and do not interact with each other, which does not reflect the real-world scenario.

(14) Link State Aware Geographic Routing Protocol (LSGR)

To evaluate the link's quality [22], an expected transmission count (ETX) metric is used. A smaller ETX value indicates a better link's quality. It helps to select quality links with high throughput, minimum transmissions, and retransmissions to deliver packets hop-by-hop to the destination. The effectiveness of the ETX routing metric is shown in [22]. However, the ETX is mainly used in proactive and opportunistic MANET's routings. The issue with using ETX in geographic routing for VANET is that it could not be adopted in a highly changing VANET environment.

The close nodes' link provides a high packet delivery rate. The ETX value of such links will be close to 1, but these links cannot contribute enough to the packet forwarding towards the destination. As a result, a trade-off situation develops between the link's reliability and forwarding towards the destination.

The paper [79] proposes an expected one-transmission advance (EOA) routing metric to enhance the greedy forwarding strategy. It modifies the greedy approach to choose a neighbor whose EOA's value is high as next-hop instead of a close neighbor. The high value of EOA means high distance coverage of packets towards a destination in one transmission. The proposed routing protocol is a link-state aware geographic routing protocol (LSGR) that modifies the ETX for the VANET environment. The EOA routing metric is

based on this modified ETX. The EOA for nodes is updated periodically. LSGR increases network throughput and reduces transmission delay. It is simulated against GPSRJ+ and GyTAR for performance evaluation.

Strong predictions in VANET cannot be made because of its dynamic characteristics. A greedy strategy is an optimization approach that needs intense care for its greedy criteria selection because the greedy approach selects the better at local with the hop of best at global. The wrong selection criteria may lead to undesirable results, so GPSR suffers from the local maximum problem. The proposed scheme uses EOA, which is based on probability.

(15) Cluster Based Routing Protocol (CBR)

The paper [80] proposes a Cluster-Based Routing (CBR) Protocol for VANETs. In CBR, the geographical area is divided into square grids, and each grid is considered a cluster head. The RSU is assumed to be a cluster head; in the absence of an RSU, a node from the network is elected as a cluster head. The data are transmitted to the destination node via neighbor cluster heads. In this way, the route discovery process will not be initiated each time when a node wants to communicate its data. As the data is forwarded to the cluster and then it is the responsibility of the cluster head to forward data packets to the destination node. It saves the memory because the routing information is not stored in every node.

It divides the geographic square area into grids, and each grid is considered a cluster, but the scheme may not work when there is an irregular area instead of the square area. The network overhead may increase when the scheme is applied to a scenario with an area like a park where the roads are at the boundaries outside the parking area. Therefore, the inner clusters with no members will be managed for no purpose. The simulation is not conducted, and it is difficult to analyze the proposed scheme's performance. The proposed scheme cannot be implemented in a pure V2V communication scenario.

**Table 8.** Routing parameters of urban scenario with traffic signals/traffic based routing protocols.

| Article | Name of the Proposed Protocol | Year of Proposal | Routing Parameters | | | | | |
|---|---|---|---|---|---|---|---|---|
| | | | MAC Protocol | Transmission Range | Operational Scenarios | Speed | No. of Nodes | Topology Size |
| [32] | P-GEDIR | 2011 | IEEE 802.11p | 200 m | Urban traffic scenario | NA | 0–200 | 2000 m * 2000 m |
| [33] | GEDDAI | 2012 | IEEE 802.11 | 200 m | Urban Mobility | 11,11.5, 12, 12.5 m/s | 500, 700, 900, 1100, 1300 | 2000 m * 2000 m |
| [34] | STAR | 2012 | IEEE 802.11b | 250 m | Urban scenario with traffic light | 20–60 km/h | 450 | 2400 m * 2400 m |
| [35] | Improved GPSR | 2012 | IEEE 802.11 DCF | 250 m | City scenario | 10–50 m/s | 100–150 | 1000 m * 1000 m |
| [36] | HyBR | 2013 | IEEE 802.11p | 300 m | Urban traffic scenario | 0–20 m/s | 20–50 | 1000 m * 1000 m |
| [37] | IAODV | 2012 | IEEE 802.11 | 250 m | City mobility model | 40 km/h and 20–50 km/h | 20–230 and 100 | 1500 m * 1500 m |
| [38] | ASAR | 2013 | IEEE 802.11 DCF | 250 m | Urban scenario with junctions | 10–50 m/s | 50–500 | 3200 m * 4000 m |
| [81] | Pro-AODV | 2015 | NA | 250 m | NA | 40 m/s | 25–250 | 1000 m * 500 m |

| | | | | | | | | |
|---|---|---|---|---|---|---|---|---|
| [40] | A Road Selection Based Routing in VANET | 2015 | NA | 500 m | City scenario with junctions | 70–90 km/h | 20–100 | 2500 m * 3000 m |
| [41] | RAGR | 2017 | IEEE 802.11b DCF | 300 m | Urban scenario | 25–50 km/h | 100–350 | 3968 m * 1251 m |
| [42] | SCRP | 2016 | NA | 250 m | Urban scenario | 30–80 km/h | 150–600 | 7500 m * 7500 m |
| [43] | JBR | 2013 | IEEE 802.11p | 250 m, 500 m and 1000 m | City scenario | 10.8–50 km/h | 300 | 1150 m * 700 m |
| [44] | LSGO | 2014 | IEEE 802.11 DCF | 250 m | Urban scenario | 10–20 m/s | 100–200 | 2500 m * 1500 m |
| [45] | GPSR-R | 2015 | IEEE 802.11 DCF | 250 m | Urban scenario | 36–108 km/h | 10 source-destination pairs, variable density | 10 km highway |
| [46] | LSGR | 2014 | IEEE 802.11 DCF | 250 m | Urban scenario | 20–80 km/h | 100–200 | 2500 m * 1500 m |
| [47] | CBR | 2010 | NA | NA | Urban scenario | NA | NA | NA |

**Table 9.** Simulation parameters of urban scenario with traffic signals/traffic based routing protocols.

| Referenced Article | Simulation Parameters/Metrics | | | | | | | |
|---|---|---|---|---|---|---|---|---|
| | Simulation Tool | Compared to | Packet Size Bytes) | Data Rate kb/s | Traffic Type | Channel Capacity | Simulation Time | Mobility Models |
| [33] | OMNeT++ | Flooding, AID and DBRS | NA | NA | NA | NA | 100 s | Urban Mobility |
| [34] | Ns-2 | VVR, GyTAR and GLF | 512 | 2 mb/s | NA | NA | NA | An urban area with traffic lights consideration |
| [32] | MATLAB | GEDIR | NA | NA | NA | NA | NA | NA |
| [35] | NS-2 and VanetMobiSim | AODV and GPSR | NA | 2 mb/s | NA | NA | 100 s | City scenario |
| [36] | NS-2 | AODV and GPRS | 1000 | 1 mb/s | NA | NA | 500 s | Urban traffic scenario |
| [37] | NS-2.34 | AODV | 512 | NA | NA | NA | 400 s | Manhattan mobility model |
| [38] | NS-2, vanetmobisim | GPSR, GSR | 512 | NA | NA | 2 mb/s | 100 s | Urban scenario with junctions |
| [40] | MATLAB | P-GEDIR, GyTAR, A-STAR and GSR | 512 | 2 mb/s | NA | NA | NA | City environment with junctions |
| [41] | NS-2.34, MOVE and SUMO | CGMR, SDR and GyTA | 512 | 3 mb/s | NA | NA | 500 s | Urban scenario |
| [42] | NS-2, MOVE and SUMO | iCAR, GyTAR and GPSR | 512 | NA | NA | NA | NA | Urban scenario |
| [43] | NS-2 | GPCR | 512 | 6 mb/s | NA | NA | 1000 s | City scenario |

| [44] | NS-2 v 2.34 | GPSRJ+ and GyTAR | 512 | NA | NA | 2 mb/s | 150 s | VanetMobiSim mobility model |
| [45] | NS-2.33 | GPSR, GPSR-L, AODV-R and MOPR-GPSR | 512 | NA | NA | 2 mb/s | 200 s | Highway scenario |
| [46] | NS-2 | GPSRJ+ and GyTAR | 512 | NA | NA | 2 mb/s | NA | VanetMobiSim |
| [47] | NA | AODV, DSDV and DSR | NA | NA | NA | NA | NA | City scenario |

**Table 10.** Performance metrics of urban scenario with traffic signals/traffic based routing protocols.

| Reference | Performance Metrics | | | | | Other Metrics |
| --- | --- | --- | --- | --- | --- | --- |
| | Packet Delivery Ratio | End to End/Average Delay | Throughput | Packet loss | Routing/Message/Communication Overhead | |
| [33] | No | Yes | No | No | Yes | Collision, coverage |
| [34] | Yes | No | No | No | No | Network latency |
| [32] | No | No | No | No | No | Avg. no. of hops, expected one-hop progress |
| [35] | Yes | No | No | No | Yes | NA |
| [36] | Yes | Yes | No | Yes | No | Normalized overhead load |
| [37] | Yes | Yes | No | Yes | No | Normalized overhead load |
| [38] | Yes | Yes | No | No | No | NA |
| [40] | No | Yes | No | No | No | N/W gap encounter, no. of hops |
| [41] | Yes | Yes | No | No | No | NA |
| [42] | Yes | Yes | No | No | No | Control overhead, control packets |
| [43] | Yes | Yes | No | No | No | NA |
| [44] | No | Yes | Yes | Yes | Yes | NA |
| [45] | Yes | Yes | Yes | No | No | NA |
| [46] | Yes | Yes | Yes | No | No | Hop count |
| [47] | Yes | Yes | No | No | No | NA |

### 4.3.2. City Map Scenario

A city map scenario is based on a city map representing the real-world graphical representation based on static information of the city [82]. The city map scenario includes streets, bus lanes, service roads, and avenues. Many protocols are designed for this scenario, and some are described below. Tables 11–13 shows routing parameters, simulation parameters, and performance metrics of city map scenarios-based routing protocols respectively.

(1) Intersection-Based Connectivity Aware Routing Icar

Intelligent Transportation System (ITS) applications are classified into one- and multi-hop applications. One hop application is generally used to forward information to their neighbor vehicles. Similarly, multi-hop applications are used to access the internet, vehicle-to-infrastructure communication, and vehicle-to-vehicle communication. For VANETs, the reliability of multi-hop applications depends heavily on effective routing.

In the design of an efficient multi-hop routing algorithm for VANET, the dynamics in topology are challenging. To handle these issues, Geographic Source-Routing GSR)[46], Car-Talk2000 [83], GEOGRAPHIC-GPSR [56], A-STAR [84], Gy-TAR [85], EGy-TAR [86], STAR [61] and NoW [87] routing protocols are proposed.

In routing schemes for VANETs, importance is given to dense highways in road selection criteria. The result is an increase in congestion because the traffic load converged to highways having high density. The node/vehicle itself is an obstacle, and the node's density may cause a communication failure. An intersection-based traffic-aware routing protocol (iCAR) is proposed in [88]. iCAR aimed to improve overall performance in a city environment using real-time traffic information and offline maps. iCAR routing choice depends on nodes/vehicles density together with average transmission delay. It evenly distributes the data packets in the VANET by ignoring the selection of dense highways with high data load as a forwarding path.

The sparse traffic in VANET multi-hop communication may lead to frequent disconnections. The simulations are supposed to validate the performance of the parameters mentioned above. Before claiming overall performance improvement, link breakage rates must be checked for this protocol. Other parameters degradation costs are not desired to improve one parameter.

(2)  Energy-Efficient Routing Using Movement Trends ERBA

Obstacle constraints in an urban environment, topology fragmentation due to high speed, roadways constrained topology, and GPS-enabled navigation are the prominent features of VANETs. Reliable, efficient, and stable routing is essential to benefit VANETs applications. In this regard, new routing protocols have been developed recently. The behavior of the drivers is critical in vehicle movement prediction. The studies have uncovered that driver behavior depends on roads, vehicle category, income, education, age, and sex [89]. It is generally believed that bus and private car drivers' behavior is different. They have different routines, routes, and speeds.

In [90], they propose an energy-efficient routing using movement trends (ERBA) based on the vehicle's mobility routine, driver behavior, and vehicle category. Public transport such as buses have their fixed routes, whereas private cars' movement is random. ERBA examines link reliability by current/motion state and distance between neighbor nodes, ensuring energy-efficient routing in VANETs. The ERBA finds enhancement in the overall routing performance by examining the driving behavior and next directions.

The proposed scheme performance is not checked against the protocol of relevant categories like Movement Prediction-based Routing (MORP). The vehicle's speed is not there in the simulation parameters. Bus speed is underestimated compared to cars, as most urban areas have dedicated signals and traffic-free lanes for buses. Energy efficiency has never been a worthy issue because vehicles are equipped with high-energy batteries and charging generators.

(3)  A Geographic Routing Protocol For Vehicular Delay-Tolerant Networks GeoSpray

The conventional routing suffers from the data delivery failure in opportunistic, partially connected, intermittent, and sparse vehicular networks because they are designed for fully connected VANETs [91] to ensure end–end connectivity along with semantics' support of existing end–end applications and transports [92]. To cover this problem, VANETs use the store-carry-and-forward (SCF) scheme. Instead, SCF does not consider path availability in the current; it assumes path availability over time. Delay tolerant networks (DTNs) utilize SCF for data delivery with maximum probability in sparsely connected VANET.

The store-carry-and-forward mechanism is used in the architecture of vehicular delay-tolerant networking (VDTN). The distinguishing feature of this architecture is the use of out-of-band signaling, data, and control plane separation and IP over VDTN. Another feature of this network is the asynchronous transmission of variable length IP packets

bundle. A control message is sent over the control plane to reserve a channel for the bundle in advance. The DTN literature has many routing protocols based on a store-carry-and-forward principle. The routing decision criteria and replication or forwarding strategy differ among these protocols. In particular scenarios, each protocol has its strength.

In the proposed GeoSpray [93] routing protocol, the store-carry-and-forward principle is used to deliver bundles. Here, the vehicle for data carriage is selected opportunistically. For routing decisions, it utilizes the information provided by positioning devices. Multi-hop routing uses multiple copy forwarding routing strategies to reduce end-end delay. The intermediate node's cheeks bundle for the clearance that they are not already delivered to the destination. GeoSpray utilizes the network resources efficiently, hence resulting in overall improved performance in terms of data delivery ratio and end–end delay.

The defined multiple-copy protocol strategy may result in an additional overhead as the intermediate nodes will check every bundle to confirm that the bundle has not been delivered. Additionally to processing overhead, it may cause delays as well. Scalability and density are two essential factors in VANET routing protocol performance, which are not considered in the simulated scenario; hence, the simulation results are not enough to confirm consistent performance.

(4)   Optimized Geographic Perimeter Stateless Routing OGPSR

Numerous routing techniques based on position are proposed in VANET. GSR was developed for a city scenario, yet did not consider the intersection. GPCR is a greedy based routing scheme that forwards the packet to a node at the intersection instead of sending it across the intersections. While GPSR and the other position based routing locating nodes using GPS are best suited for VANET. Thus, various enhancements to this strategy are proposed. Greedy Perimeter Stateless Routing with Movement Awareness (GPSR-MA) [94] takes into consideration speed, distance, node movement in making route decisions. Another improvement to GPSR is proposed in [95] and uses a formula that determines the forwarding node on the basis of triangular area and distance of the relay. Moving Directional Based Greedy (MDBG) [96] routing resolves the direction problem in greedy schemes. It uses destination requests, destination replies, and hello messages to determine the direction of the nodes. At [64], the proposed technique picks an efficient path using the hello packet. It resolves the problem of local maxima in GPSR, but the delay is not considered here.

In greedy schemes, the optimized GPSR [97] proposal solves the problem to guarantee the right selection in the appropriate direction. The criteria of greed in GPSR is finding the proceeding node on the basis of distance towards the destination. Therefore, there is a possibility of wrong selection in the wrong direction. To avoid this, an additional parameter of direction is included in the selection criteria. To select the forwarding node in the correct direction, OGPSR employs the arc tangent rule. For vertical and horizontal measurements having two lanes each, the arc of the tangent is further used to improve the greedy forwarding.

The authors discuss MDBG, GPSR-MA and other improved schemes of GPSR in the related work. The proposed system will also be tested for its performance against these improvements. The transmission range of the nodes is not specified in the parameters, which makes it difficult to be analyzed for the outcomes of certain parameters. The performance improvements in the city scenario are not remarkable.

(5)   A VANET Routing Based on The Real-Time Road Vehicle Density

GPSR is a geographical-based routing protocol, and many of the recently proposed routing schemes are based on it. GPSR utilizes immediate neighbor node information for its greedy decisions to forward packets. In a region where the proposed greedy procedure of GPSR cannot forward packets, the packets are then forwarded along the region's perimeter. The driving habits, vehicle density, and high mobility challenges affect the performance of GPSR in VANET. In a city environment, road layouts define the topology of

the VANET. The packets are forwarded to the destination using vehicles on the roads. In this scenario, the performance of GPSR is not effective, as the greedy procedure may forward the packets to a low-density region in the greed of the shortest path.

To overcome this issue, VANET routing based on the real-time road vehicle density in a city environment is proposed [98]. It provides stable routing for V2V communication in a city environment by considering high vehicle density regions for route establishment. The vehicle density is measured by beacon messages and a road information table. The source node can select a stable path for packet forwarding as it is aware of vehicle density on the roads—the stable path to the destination results in minimum transmission delay.

The proposed scheme does not consider the direction of forwarding nodes, which may lead to long path selection causing transmission delay. The authors of the proposed scheme claim minimum transmission delay due to stable path selection in their routing scheme, but the scheme is not checked for transmission delay. Secondly, they claim that the proposed scheme performs better than other proposed variations to GPSR, such as CAR, A-STAR, VADD, and SADV, whereas the claim is not validated through simulation. The proposed scheme is validated through simulation for packet delivery ratio and routing overhead against GPSR.

**Table 11.** Routing parameters of city map scenario-based routing protocols.

| Article | Name of the Proposed Protocol | Year of Proposal | Routing Parameters | | | | | |
|---------|---|---|---|---|---|---|---|---|
| | | | MAC Protocol | Transmission Range | Operational Scenarios | Speed | No. of Nodes | Topology Size |
| [48] | ERBA) | 2013 | NA | 500 m | Real urban scenario | NA | 50–150 buses and 70–300 cars | 1.5 km * 2 km |
| [49] | GeoSpray) | 2011 | NA | 30m Omni directional | City map scenario | Variable speed with avg speed = 50 km/h | 100 | 4500 m * 3400 m |
| [50] | OGPSR | 2016 | IEEE 802.11 | NA | Urban map scenario | 10–20 m/s | 50–125 | 500 m * 500 m |
| [51] | A VANET routing based on the real-time road vehicle density | 2013 | IEEE 802.11 | 250 m | City scenario | 20–55 km/h | 150–300 | 3000 m * 3000 m |
| [52] | iCAR) | 2013 | NA | 250 m | City map | NA | 6–12 per lane/km | 7000 * 7000 m² |

**Table 12.** Simulation parameters of city map scenario-based routing protocols.

| Referenced Article | Simulation Parameters/Metrics | | | | | | | |
|---|---|---|---|---|---|---|---|---|
| | Simulation Tool | Compared to | Packet Size Bytes) | Data Rate kb/s) | Traffic Type | Channel Capacity | Simulation Time | Mobility Models |
| [48] | Ns-2.34 | ROMSGB, AODV | 512 | 128 | NA | NA | 30 min | Real urban constrained mobility |
| [52] | Mat lab | GPSR, GyTAR | 512 | 12 mb/s | NA | NA | NA | NA |
| [49] | Opportunistic Network Environment ONE) | Epidemic, Spray and Wait, | variable | NA | NA | 4.5 mb/s | 6 hrs | City map scenario |

| | simulator called VDTN-sim | PRoPHET and GeOpps | | | | | | |
|---|---|---|---|---|---|---|---|---|
| [51] | NS-2 | GPSR | NA | 20-40 | NA | NA | 200 sec | City environment |
| [50] | NS-2 | GPSR | 512 | NA | NA | NA | 100 sec | Urban map scenario |

**Table 13.** Performance metrics of city map scenario-based routing protocols.

| Reference | Performance Metrics | | | | | |
|---|---|---|---|---|---|---|
| | Packet delivery Ratio | End to End Delay/Average Delay | Throughput | Packet Loss | Routing/Message/Communication Overhead | Other Metrics |
| [48] | Yes | Yes | No | No | Yes | NA |
| [52] | Yes | Yes | No | No | Yes | NA |
| [49] | No | No | No | Yes | Yes | No. of initiated bundles |
| [51] | Yes | No | No | No | Yes | NA |
| [50] | No | Yes | Yes | No | Yes | NA |

### 4.3.3. Urban Scenario with Streets

This scenario is also urban-based, explicitly considering the street's environment. The road vehicles have obstacles in-between, although they remain close to each other. The restricted movement of vehicles affects the routing behavior of VANET routing protocols. Tables 14–16 shows routing parameters, simulation parameters, and performance metrics of urban scenarios with streets based routing protocols respectively.

(1) Automatic Tuned Optimized Link State Routing

Mobile ad hoc networks (MANET) routing protocols are unsuitable for VANET due to high mobility in VANET. Optimized link state routing (OLSR) is a renowned routing protocol in MANET. OLSR is still used in VANET deployment because of its adaptability to the change in topology [98,99]. In this deployment, the congestion issue arises because of routing control packets traffic.

In VANET, high mobility with limited Wi-Fi coverage leads to rapid changes in topology and fragmentation issues. In such a scenario, routing data packets is challenging. An efficient routing strategy is decisive in VANETs. Toutouh, J., J. GarcÃ­a-Nieto, and E. Alba in [100] find the optimal configuration of the OLSR parameters by utilizing different optimization techniques. The automatic OLSR is then checked for performance under realistic VANET scenarios of the city.

Toutouh, J., J. GarcÃ­a-Nieto, and E. Alba in [100] Uses SA, DE, GA, PSO, RAND, and RFC heuristic algorithms to optimize the parameters for OLSR offline. OLSR parameters are optimized for three different scenarios. The results show that one parameter is optimized by one algorithm, and another algorithm optimizes another parameter. It means one optimization technique cannot optimize all the parameters of the OLSR. This behavior is different for different scenarios. Secondly, although the overhead optimization process is offline, the scenarios must be pre-defined. If the vehicle attains a different scenario at run time, then the OLSR behavior is not defined. The author says that the automatic tuned OLSR is compared for performance with the standard one as in RFC 3626 OLSR and human experts from state of the art. Still, the simulation results only show OLSR parameters' optimization using different heuristic algorithms. A simulation for performance comparison with other protocols is missing.

(2) Mobility Aware Zone Based Ant Colony MAZACORNET

VANET routing schemes can be recognized as a single path, carry and forward path, or multipath routing. Ad hoc On-Demand Multipath Distance Vector (AOMDV) [101], S-AOMDV [52] and AODVM [102] multipath routing schemes are the enhanced versions of AODV. These are non-scalable re-active schemes. S-AOMDV needs extra messages to enhance route finding, and route failure may result in traffic congestion and bandwidth wastage.

Numerous research in MANET [103,104] validated that bio-inspired algorithms such as ant colony optimization (ACO) can be used magnificently to design an efficient routing algorithm. These schemes are more advantageous than other routing schemes [100,105].

The information is shared locally to minimize the control message overhead for upcoming routing choices. If a link fails on the former selected route, these schemes find other paths allowing us to choose another path.

Mobility Aware Zone-based Ant Colony Optimization Routing for VANET (MAZA-CORNET) is a hybrid scheme [105] that was proposed in [106]. It divides the network nodes into zones to efficiently utilize the bandwidth. MAZACORNET used a proactive method for intra-zone communication and a reactive method for inter-zone communication to identify routes. The congestion and broadcast messages are reduced because it uses native information stored in each zone. The vehicle's mobility pattern, degree, speed, and fading conditions are used to design a multipath routing scheme.

The authors of MAZACORNET claim that the mobility-aware ant colony optimization routing algorithm for vehicular ad hoc networks (MAR-DYMO) [107] is the only nature-inspired algorithm proposed for VANET. Still, its performance is not compared with the proposed scheme. The suggested scheme is near cluster-based routing and is not compared with the current cluster-based schemes. The velocity and communication range of vehicles is not stated.

(3) Connectionless Approach for Vehicular Ad Hoc Networks in Metropolitan Environment Came

Many geographically routing techniques proposed for VANETs create a source to destination route. In these connection-oriented protocols, there is only one data transmission route. The single established route may be interrupted due to the low density of vehicles. More control messages must be sent by the protocol to restore the inactive route, which may result in an end-to-end delay. The solution to these problems is proposed in [54,108] in the form of multipath routing protocols. Again, the transmission of control packets is a problem. Connectionless routing protocols are therefore proposed [109,110], where no route needs to be established for the transmission of data. Relay nodes are chosen depending on topology changes and mobility of vehicles, but even for these routing protocols, the average end-to-end delay needs to be improved.

In [111], the author proposes a connection-less approach for VANETs in a metropolitan scenario called CAME. Based on changes in the topology, different packet delivery strategies are used in the proposed scheme, and it does not require a route to be specified in advance. There are different routing strategies in this scheme for roads and junctions. A reference line is developed by it to support the selection of the relay node and then the onward relays to the destination. Similarly, source and destination nodes communicate with each other. Likewise, it takes into account the flow of data and avoids congestion and disconnections for assurance of the packet's delivery. Thus, the time delay is minimized and the ratio of packet delivery is increased with minimal control overhead.

However, extra computational overhead in the proposed system for mode selection and location determination. This repeated process to select the next relay may result in time delay. The average number of hops used in data transmission is an important factor to consider.

**Table 14.** Routing parameters of urban scenarios with streets-based routing protocols.

| Article | Name of the Proposed Protocol | Year of Proposal | Routing Parameters | | | | | |
| --- | --- | --- | --- | --- | --- | --- | --- | --- |
| | | | MAC Protocol | Transmission Range | Operational Scenarios | Speed | No. of Nodes | Topology Size |
| [53] | Automatic tuned OLSR | 2012 | IEEE 802.11b | 250 m | Realistic city | 10–50 km/h | 30 | 1200 * 1200 m2 |
| [54] | MAZACOR-NET) | 2013 | IEEE 802.11b and 802.11p | NA | Urban traffic scenario with streets | NA | 25, 50, 75 and 100 | 500 m * 500 m |

| [55] | CAME) | 2016 | IEEE 802.11p | 300 m | Metropolitan environment | 0–20 m/s | 5–25 per 100 m | 1000 m * 1000 m |

**Table 15.** Routing parameters of urban scenario with streets based routing protocols.

| Referenced Article | Simulation Parameters/Metrics | | | | | | | |
|---|---|---|---|---|---|---|---|---|
| | Simulation Tool | Compared to | Packet Size Bytes) | Data Rate kb/s) | Traffic Type | Channel Capacity | Simulation Time | Mobility Models |
| [53] | Ns-2 | NA | NA | NA | NA | 6 mb/s | NA | NA |
| [54] | NS2, VanetMo-biSim and AWK | AODV, AMODV and GPSR | 512 | NA | NA | NA | 2000 sec | NA |
| [55] | NS-2, MOVE | WPB, CLA-S | 512 | NA | NA | NA | 100 sec | Metropolitan En-vironments |

**Table 16.** Performance metrics of urban scenario with streets based routing protocols.

| Reference | Performance Metrics | | | | | |
|---|---|---|---|---|---|---|
| | Packet Delivery Ratio | End to End De-lay/Average Delay | Throughput | Packet Loss | Routing/Mes-sage/Communica-tion Overhead | Other Metrics |
| [53] | Yes | Yes | No | Yes | No | Normalized routing load |
| [54] | Yes | Yes | No | No | Yes | NA |
| [55] | Yes | Yes | No | Yes | No | Control over-head, control packets |

*4.4. Grid Based Scenario*

The grids display the perceived system's axis [112]. The grid-based scenario represents the grid of road lanes that intersect each other. The designing considerations of routing protocols for this kind of scenario differ from different described scenarios. Tables 17–19 show routing parameters, simulation parameters, and performance metrics of grid-based scenarios routing protocols respectively.

4.4.1. Grid-Based Predictive Geographical Routing GPGR

Vehicle-to-vehicle communication is a multi-hop communication between vehicles with wireless connectivity and without some static infrastructure [113]. When a node wants to communicate with another node in VANET, the relay nodes transmit the data packets to the destination. The quick topology changes and other features of VANET, such as the reserved movement of vehicles due to obstacles, and traffic signals in cities, cause frequent link breakage [114]. Hence, existing MANET routing schemes are not appropriate for VANET. Topographical routing scheme such as Greedy Perimeter Stat less Routing GPSR) is more relevant in such scenarios [115]. Compared to predestination routing entries, topographical forwarding only possesses information about their neighbors.

The problem with these topographical routing schemes is the local maxima triggered during the relay node selection scheme, resulting in selecting the nearest node to the destination node as a relay [116]. The GPGR scheme was proposed to handle these issues. GPGR divides the roads into a two-dimensional grid using the map. GPGR uses a road grid throughout the relay selection procedure and predicts the next geographic location of the vehicle considering all probable node travels. The next geographic location of the vehicle can be predicated, and an optimum relay vehicle is selected. NS2 was used for

simulation, and the performance was tested in terms of link breakage and delivery ratio with the possibility of local maxima.

The routing scheme discussed above was based on a single/specific scenario, and a high-speed VANET with dynamic topology features may operate in a different environment. Two performance metrics were assumed during simulation, i.e., packet delivery and link breakage ratio, but the processing overhead, latency, and delay should be carefully checked to validate the performance. A VANET deployed for real-time applications may increase latency using the earlier routing scheme.

### 4.4.2. Reliable Inter-Vehicular Routing River

Performance in VANET routing is closely concerned with the availability of network nodes for packet forwarding. No. of nodes per unit area density varies with the traffic signals and stop signs, etc., in urban areas. In such a scenario, VANET forwards the packets along the streets. The presence of a vehicle in the street is not always ensured as the real-world scenario is not predictable and uniform. Due to construction, traffic rules, and events, the situation varies with time, date, weather, and diversions. Routing protocols like GSR and SAR [117] ignore these traffic considerations. A-STAR design considers the dense vehicular scenario of bus schedules that is based on static traffic information. The network is updated periodically against real-world changes. Network link breakage may occur due to network gap, sparse density, and low transmission range of nodes. The change in network density is dynamic and cannot be covered with static information. In STAR, routing decisions are based on relative density around a vehicle. CAR [118] examines neighbors' density and controls beaconing to the neighbors to avoid congestion in a dense network. SADV [119] determine densities by analyzing packet delay. Using an offline map ACAR [115] divides the area into clusters and measures its connectivity probability. VADD [120] adopts a carry-and-forward approach to cover disconnection in the network, which causes delay.

Bernsen, J. and D. Manivannan in [121] propose a position-based routing protocol named Reliable Inter-Vehicular Routing (RIVER). It monitors traffic and uses a greedy strategy to forward packets on a determined, reliable route. Real-time traffic monitoring is achieved by continuously transmitting probe messages in streets and examining adjacent intersection communication. Instead of traditional network flooding or broadcasting that causes congestion, RIVER determines route reliability by beacons, probes, and piggybacking the route reliability data on routing messages. It recalculates the route dynamically at any point as the message leaves the knowledge zone. The same procedure is adopted for route recovery in case of link breakage.

The proposed routing scheme forwards messages on a route using a greedy approach; if the selected path is not the shortest one, then a delay in delivery may occur. The scheme performance is shown for a dense city environment and will not be as effective for other scenarios. The packet delay is not measured in the simulation and may be greater than the STAR, GPSR, and shortest path routing, as the number of hops in the proposed scheme are comparatively high.

### 4.4.3. An Efficient Prediction-Based Forwarding Strategy

In VANET communication, if the distance between source and destination is high, the number of relay nodes will be high. This may affect the packet delivery ratio in highly mobile VANET due to link instability. The solution to this issue is addressed in different research works on the cost of bandwidth resources and network capacity [41]. It affects the VANET QoS efficiency. The improvement in path stability, lifetime, and minimum impact of link breakage on data dissemination are the important design parameters that can ensure reliable routing in VANET [32,122]. The minimum distance and relative mobility between the two nodes lead to link robustness. A series of short links in a path makes it more reliable with a low chance of breakage, but it increases the end-to-end delay and

reduces bandwidth efficiency. To resolve this trade-off, link lifetime prediction is a possible solution. Roads and speed limits restrict mobility in VANET. The information can be obtained from these predefine paths and speed limits to predict link duration.

The authors in [122] suggest a data forwarding strategy based on link duration prediction. It solves the trade-off between link reliability and data transmission efficiency in VANETs. The location and speed information is appended with periodic control packets in existing inter-vehicle interactions to predict the link duration. The proposed scheme uses the measured impact of velocity and distance over data transmission efficiency to optimize the forwarding path selection. The proposed scheme introduces two-hop neighbor information to improve the path selection process further to select a reliable path. The proposed routing strategy can be used in any routing protocol for performance improvement.

The proposed scheme is based on link duration predictions that increase the computational overhead and may lead to high end-to-end delay. The proposed scheme is not compared for performance against other relevant routing protocols. Along with location and speed, change in direction at junctions is another important parameter for link reliability that is not considered in the proposed strategy.

**Table 17.** Routing parameters of grid scenario-based routing protocols.

| Article | Name of the Proposed Protocol | Year of Proposal | Routing Parameters | | | | | |
|---|---|---|---|---|---|---|---|---|
| | | | MAC Protocol | Transmission Range | Operational Scenarios | Speed | No. of Nodes | Topology Size |
| [56] | RIVER) | 2012 | NA | NA | 5 streets horizontal and vertical 400 apart | 11–51 km/h | 100–300 | 6.05 km * 6.05 km |
| [57] | GPGR) | 2012 | IEEE 802.11 | 125 m | Urban with traffic signals | 0–80 km/h | 100–200 | 700 * 1000 m² |
| [58] | An Efficient Prediction-Based Forwarding Strategy | 2015 | IEEE 802.11p | 200 m | Roads grid scenario | 5–20 m/s | 30 per km | 1000 m * 1000 m |

**Table 18.** Simulation parameters of urban scenario with streets-based routing protocols.

| Referenced Article | Simulation Parameters/Metrics | | | | | | | |
|---|---|---|---|---|---|---|---|---|
| | Simulation Tool | Compared to | Packet Size Bytes) | Data Rate kb/s) | Traffic Type | Channel Capacity | Simulation Time | Mobility Models |
| [57] | Ns-2 | GPSR, GPUR, GPCR | 1000 | NA | NA | 2 mb/s | NA | NA |
| [56] | NS-2 | STAR, GPSR and shortest-path VANET routing | 512 | 4 | NA | NA | 200 sec | Streets scenario |
| [58] | NA | NA | 1024 | 2 mb/s | NA | NA | 4800 sec | Manhattan mobility model |

**Table 19.** Performance metrics of urban scenario with streets-based routing protocols.

| Reference | Performance Metrics |
|---|---|

| | Packet Delivery Ratio | End to End Delay/Average Delay | Throughput | Packet Loss | Routing/Message/Communication Overhead | Other Metrics |
|---|---|---|---|---|---|---|
| [57] | Yes | No | No | No | No | Link breakage rate |
| [56] | No | No | Yes | No | No | Forward per rout, rout transmit time |
| [58] | Yes | Yes | No | No | No | Average no. of hops |

## 5. Modern Vanets Schemes

The modern techniques used for solving the issues of VANETs are discussed in this section.

A. Software-Defined Networking

Software-Defined Networking (SDN) subdivides the communication protocol functions into two modules, namely communication policies or routing decisions and data forwarding. Policies in SDNs are centrally controlled, and policies are typically implemented on fixed infrastructure/roadside units (RSUs) and not on mobile devices. These mobile devices forward the data in accordance with the centrally defined policies. Open-Flow [123,124] is the known protocol used for central control in SDN.

B. Named Data Networking (NDN)

Data content in NDN is named and not the end-to-end devices. The consumer in NDN sends the interest packets, and the provider of the contents forwards the content data after receiving the interest packets on the route where the interest packets were received again. In VANET, this can serve various applications depending on the interests of the consumers [125,126].

## 6. Conclusions and Future Work

The literature study of different VANET routing protocols shows that scenario-based routing is another categorization of VANETs routing protocols besides the design strategy. This categorization not only provides an additional aspect for studying the VANETs routing protocols, but also reveals that the requirements of routings are different for different scenarios. Since these facts, we have developed a novel taxonomy for the VANET routing schemes to facilitate the young researchers to study the schemes from their domain as literature. However, the limitation of this study is that it only covers the conventional routing schemes. In future studies, this categorization can be further improved by keeping all this taxonomy in one category and the newly emerging schemes in other categories. Then, the fact sheet is derived from the survey to show that different routing and simulation parameters are considered for different scenarios, and the protocols are checked for different performance metrics. In other words, a routing protocol designed for one scenario can be only suitable for that scenario, and its performance in another scenario is not guaranteed. This finding led us to the development of a dynamic routing protocol in future that ensures the consistent performance throughout the operation regardless of the underlying operational scenario. The researchers working on routing protocols in VANET are supposed to agree on a unified standard that can ensure the quality of service throughout the vehicular ad hoc network operation in the Intelligent Transformation System. It is shown in the literature that a particular routing protocol cannot perform optimally in all scenarios or mobility models (MM), but can outperform another protocol in the new topology and mobility model. In the future, a dynamic routing protocol is required for the dynamics in topology, mobility models, and network performance met-

rics. A reactive supervisory protocol is intelligent enough to identify the operational environment and invoke such a protocol for further communication that is best for the identified operating environment.

**Author Contributions:** I.W. and S.T.; methodology, I.W.; software, I.W., M.A.; validation, I.W., F.U.; formal analysis, I.W., F.U.; investigation, M.K.; resources, M.K., F.U.; data curation, I.W.; writing—original draft preparation, I.W., F.U., M.K.; writing—review and editing, M.A.; visualization, F.U., M.K.; supervision, S.T.; project administration, S.T..; funding acquisition, A.S.A.; S.S.A. All authors have read and agreed to the published version of the manuscript.

**Funding:** Taif University Researchers supporting Project number (TURSP-2020/311): Taif University, Taif, Saudi Arabia.

**Acknowledgments:** Taif University Researchers supporting Project number (TURSP-2020/311): Taif University, Taif, Saudi Arabia.

**Conflicts of Interest:** The authors declare no conflict of interest.

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
