# Peer review of "Vehicular Ad Hoc Networks Routing Strategies for Intelligent Transportation System"

_electronics, doi:10.3390/electronics11152298_

Round 1

Reviewer 1 Report

My concerns are as follows.

-          The novelty of the work is not clear. Authors should introduce the novelty of the work compared with the existing proposals. There are many survey articles in the same context; authors should illustrate how their work differs from these existing proposals.

-          The main contributions of the article should be clearly presented at the end of the introduction section.

-          Distributed edge computing is a recent paradigm used for VANETs; authors should provide the role of multiple access edge computing and fog computing in routing schemes of VANET.

-          Also, AI and ML are commonly used for routing of VANETs; authors should provide a section for such context.

-          The challenges and the current research directions should be added to the review.

Author Response

Dear Sir,

Please find the attached replies to comments of reviewers and updated paper. Thank You

Reviewer 2 Report

The main motivation of this study must be clarified in the introduction section to facilitate future readers. And then they can find the main idea and how the given problem has been solved. Try to put your contribution in a challenge and solution manner, in which you show the problem of the existing studies and your solution which is the new contribution point.

In my opinion, the introduction should be further improved to better specify the motivation of the work and the characteristics of the methodology. As it is, it is not clear how it works. Additionally, the author should state clearly in which aspect this work extends the state of the art, i.e., what is the novelty? Consider https://www.mdpi.com/2079-9292/10/18/2250. About the literature review. Each paper should clearly specify what is the proposed methodology, novelty, and results with experimentation. At the end of related works, highlight in some lines what overall technical gaps are observed in existing works, that led to the design of the proposed approach. To better delineate the context and the different possible solutions, you can consider https://ieeexplore.ieee.org/abstract/document/9409962.

The authors should focus on their unique work and contributions at first, and they should support their conclusion with numerical results. Then, the limitations of this paper should be discussed. Accordingly, the future work of this paper can be drawn.

Author Response

Dear Sir,

Please find the attached replies to the comments of Reviewers and updated manuscript. Thank You

Round 2

Reviewer 2 Report

The authors improve the paper, but suggestions provided on point 2 and 3 have to be referenced. The conclusion should be improved. The authors should focus on their unique work and contributions at first, and they should support their conclusion with numerical results. Then, the limitations of this paper should be discussed. Accordingly, the future work of this paper can be drawn.

Author Response

Please find the attached comments to the reviewers.
